# Transglutaminase 2 contributes to a TP53-induced autophagy program to prevent oncogenic transformation

**Shi Yun Yeo[1†], Yoko Itahana[1†], Alvin Kunyao Guo[1], Rachel Han[1], Kozue Iwamoto[1], Hung Thanh Nguyen[1], Yi Bao[1], Kai Kleiber[1], Ya Jun Wu[2], Boon Huat Bay[2], Mathijs Voorhoeve[1*], Koji Itahana[1*]**

[1]Cancer & Stem Cell Biology Program, Duke-NUS Medical School, Singapore; [2]Department of Anatomy, Yong Loo Lin School of Medicine, National University Health System, Singapore

**Abstract** Genetic alterations which impair the function of the TP53 signaling pathway in *TP53* wild-type human tumors remain elusive. To identify new components of this pathway, we performed a screen for genes whose loss-of-function debilitated TP53 signaling and enabled oncogenic transformation of human mammary epithelial cells. We identified transglutaminase 2 (TGM2) as a putative tumor suppressor in the TP53 pathway. TGM2 suppressed colony formation in soft agar and tumor formation in a xenograft mouse model. The depletion of growth supplements induced both *TGM2* expression and autophagy in a TP53-dependent manner, and TGM2 promoted autophagic flux by enhancing autophagic protein degradation and autolysosome clearance. Reduced expression of both *CDKN1A*, which regulates the cell cycle downstream of TP53, and *TGM2* synergized to promote oncogenic transformation. Our findings suggest that TGM2-mediated autophagy and CDKN1A-mediated cell cycle arrest are two important barriers in the TP53 pathway that prevent oncogenic transformation.

*For correspondence: mathijs. voorhoeve@gmail.com (MV); koji. itahana@duke-nus.edu.sg (KIt)

[†]These authors contributed equally to this work

**Competing interests:** The author declares that no competing interests exist.

## Introduction

The TP53 (known as p53) protein is a central component of the tumor suppressive network that monitors oncogenic transformation (*Vogelstein et al., 2000*). Activation of TP53 by oncogenic stress stimulates the transcription of a host of genes, in particular those involved in cell cycle arrest, apoptosis, metabolism, and autophagy (*Bieging and Attardi, 2012*). The relative contribution of these genes to tumor suppression by TP53 is likely to be tissue- and context-dependent, but it is clear that they play complementary roles. For example, mutations in *CDKN1A* (known as *p21*), which is one of the best-characterized direct target genes of TP53 and prevents cell cycle progression (*Abbas and Dutta, 2009*), are much less frequent than mutations in *TP53*. Moreover, *Cdkn1a* knockout mice have a much lower tumor penetrance than *TP53* knockout mice (*Martin-Caballero et al., 2001*), suggesting that additional TP53 targets must contribute to tumor suppression (*Brady et al., 2011*).

It has been shown that TP53 activity is required to prevent tumorigenesis in vivo (*Bieging and Attardi, 2012*) and transformation in vitro (*Hahn et al., 1999*). For example, primary human mammary epithelial cells (HMECs) can be fully transformed to form colonies in soft agar and tumors in immunocompromised mice by overexpressing TERT, HRAS[V12], and the SV40 oncoproteins large T and small T, which inactivate TP53 and RB1/pRB, and PP2A, respectively (*Elenbaas et al., 2001*; *Hahn et al., 2002*). This in vitro transformation model is particularly powerful for identifying and studying putative tumor suppressor genes in the TP53 pathway (*Drost et al., 2010*; *Voorhoeve et al., 2006*), especially compared to cancer-derived cell lines or spontaneously

**eLife digest** Cancers grow from rogue cells that manage to defy the strict rules that normally stop a cell from dividing when it should not. Each cell contains many proteins that are responsible for implementing these rules, and thus help to prevent tumors from forming. One of these proteins – p53 (which is also called TP53) – plays a central role in this process. Information about many processes within and around a cell filters through the p53 protein, before being passed on to a range of different proteins.

The proteins that are alerted by p53 are commonly referred to as its 'downstream effectors', and it is these proteins that stop cells from dividing too much. For example, the protein p21 (also called CDKN1A) – which is the best understood of p53's downstream effectors – hinders the machinery that causes cells to divide. Other p53 effectors can cause cells to kill themselves to prevent cancer growth. However, recent experiments with mice predicted that there may be other p53's effectors that are important too.

Yeo, Itahana et al. have now depleted the proteins that potentially work in p53's network, one by one, in human cells called mammary epithelial cells, to test if these cells can become cancerous in the laboratory. The experiments showed that another downstream effector protein of p53 – an enzyme called transglutaminase 2 – contributes to prevent these mammary epithelial cells from becoming cancerous. Transglutaminase 2 promotes a process known as autophagy, which recycles damaged and old components of the cell, and therefore normally helps to keep cells healthy.

Yeo, Itahana et al. also demonstrated that the effects of both p21 and transglutaminase 2 are critical to stop human mammary epithelial cells grown in the laboratory from dividing too much and from forming tumors when injected into mice.

These experiments provide a deeper understanding of how most cells manage to remain healthy rather than becoming cancerous and reveal a potential new target for the early detection of cancer. Further investigations could now explore whether therapies could re-activate this enzyme to prevent or treat cancer.

immortalized cells such as MCF10A cells in which the tumor suppressive network has been inactivated in a variety of ways (*Kadota et al., 2010*).

Given the crucial role of the TP53 pathway in tumor suppression, the significant proportion of tumors that still express wild-type *TP53* are likely to harbor alternative lesions that override TP53 activity, most prominently MDM2 overexpression or loss of CDKN2A (p14ARF) expression (*Vogelstein et al., 2000*). In addition, a significant number of *TP53* wild-type breast cancer tumor lose expression of BRD7, a transcriptional cofactor of TP53, compared to *TP53* mutant tumors (*Drost et al., 2010*; *Miller et al., 2005*). Therefore, to identify genes that modulate the TP53 pathway for tumor suppression, we developed a loss-of-function screen employing HMECs. In HMECs, the TP53 pathway is intact, but the RB1/pRB pathway is disrupted due to silencing of the *CDKN2A (INK4A/p16)* promoter (*Stampfer and Yaswen, 2000*). Utilizing these characteristics, we established a primary HMEC malignant transformation system that is genetically defined to depend on the loss of TP53 activity for full transformation, which can be assessed by analyzing colony formation in a soft agar assay.

This screen uncovered tissue transglutaminase 2 (*TGM2*) as a tumor suppressor that inhibits oncogenic transformation of HMECs. We showed that *TGM2* expression is regulated by TP53 to suppress oncogenic transformation of, and tumor formation by, primary HMECs. We provide evidence that reduced *TGM2* expression induces colony formation in soft agar possibly due to defects in autophagy, specifically autophagic protein degradation and autolysosome clearance. Importantly, simultaneous knockdown of *TGM2* and *CDKN1A* synergistically promotes transformation, revealing the complementary and essential roles of TP53-induced autophagy and cell cycle arrest in tumor suppression.

## Results

### TGM2 suppresses oncogenic transformation of primary human mammary epithelial cells

To identify new genes within the TP53 tumor suppressor pathway, we established an assay in which the loss of TP53 signaling promotes oncogenic transformation. We employed human mammary epithelial cells (HMECs) since the TP53 pathway is intact, but the RB1/pRb pathway is disrupted due to silencing of the *CDKN2A (INK4A*/p16) promoter (*Stampfer and Yaswen, 2000*). HMECs require TERT, oncogenic ER-HRAS[V12], SV40 large T and small T antigen for full transformation (*Hahn et al., 2002*). However, we did not use large T antigen since it would perturb the TP53 pathway. ER-HRAS[V12] is a fusion protein of HRAS[V12] with the hormone-binding domain of the estrogen receptor, and is activated by the addition of 4-Hydroxy-Tamoxifen (4-OHT) (*Voorhoeve et al., 2006*). This inducible HRAS system allowed us to minimize the emergence of aberrant clones arising from HRAS oncogenic stress. These cells are referred to as HMEC[TERT/ST/ER-RasV12] cells throughout this paper.

To develop a soft agar screen that suppresses colony formation in *TP53* wild-type but not *TP53* depleted cells, we first plated HMEC[TERT/ST/ER-RasV12] cells in medium supplemented with 4-OHT (to activate HRAS[V12]), EGF, insulin, and hydrocortisone (*Drost et al., 2010*; *Hahn et al., 2002*). Unexpectedly, many colonies grew in soft agar under these conditions, even though the TP53 pathway was not specifically inhibited (*Figure 1—figure supplement 1*, first column). In addition, the number of colonies was not significantly increased by *TP53* shRNA (*Voorhoeve and Agami, 2003*) (*Figure 1—figure supplements 1* and *2*), suggesting that TP53 activity does not inhibit oncogenic transformation under these conditions. Therefore, we tested more stringent conditions that would avoid transformation due to potentially oversaturated growth supplements. We found that HMEC[TERT/ST/ER-RasV12] cells produced significantly fewer colonies when they were grown in medium with only 4-OHT for the first 3 days, followed by medium with 4-OHT, EGF, insulin, and hydrocortisone (*Figure 1A*, first column). Importantly, knockdown of *TP53* substantially increased the number of colonies, suggesting that the loss of TP53 activity is required for transformation under these conditions (*Figure 1A* and *Figure 1—figure supplement 3*). Therefore, we used these conditions to identify genes whose loss compromises the TP53 pathway.

We searched a publicly available breast cancer expression array dataset for genes with reduced expression in a significant number of *TP53* wild-type tumor samples compared to *TP53* mutant tumor samples (GSE3494) (*Figure 1B* and *Supplementary file 1*) (*Miller et al., 2005*). We reasoned that there is selective pressure to abrogate TP53 signaling during carcinogenesis, and that the loss of expression of TP53 pathway components would be more frequent in a subset of *TP53* wild-type tumors compared to *TP53* mutant tumors. Thus, genes with reduced expression in a subset of *TP53* wild-type (see red circle in *Figure 1—figure supplements 4,5,* and *6*) but not in mutant *TP53* tumors are potential members of the TP53 pathway. We selected 122 candidates with significantly lower expression in a subset of *TP53* wild-type tumor samples compared to *TP53* mutant tumor samples using Chi-square analysis (*Figure 1B* and *Supplementary file 1*).

We individually transduced shRNAs for each of these 122 candidate genes into HMEC[TERT/ST/ER-RasV12] cells, and evaluated the effect of knockdown on colony formation in soft agar assays (*Figure 1B* and *Supplementary file 1*). *CDKN1A shRNA* (*Voorhoeve et al., 2006*) was used as a positive control, since it is a well-known downstream target gene of TP53 and since reduced *CDKN1A* expression promotes cell transformation (*Schaefer et al., 2010*; *Zhang et al., 2013*; *Zou et al., 2002*). Among the 122 candidate genes, there were four shRNAs that produced colonies in soft agar from the primary screen (*Supplementary file 1*). To exclude off-target effects, we constructed additional shRNAs against the four genes in a secondary screen (*Supplementary file 2*), and found that knockdown of only one, *TGM2*, produced colonies in soft agar with at least two independent shRNAs (denoted as *TGM2#1* and *TGM2#2*) (*Figure 1C* and *Figure 1—figure supplements 7* and *8*). The number of colonies formed correlated with the efficiency of *TGM2* knockdown (*Figure 1D and E*). To exclude the possibility that these two independent shRNAs against *TGM2* share off-target activity, we restored *TGM2* expression in HMEC[TERT/ST/ER-RasV12] cells expressing *TGM2*#1 shRNAs with an shRNA-resistant cDNA (*TGM2[R]*). *TGM2* restoration at physiological levels significantly suppressed colony formation (*Figure 1F and G*). Taken together, these findings uncover *TGM2* as a putative tumor suppressor gene that functions within the TP53 pathway to prevent oncogenic transformation of HMECs.

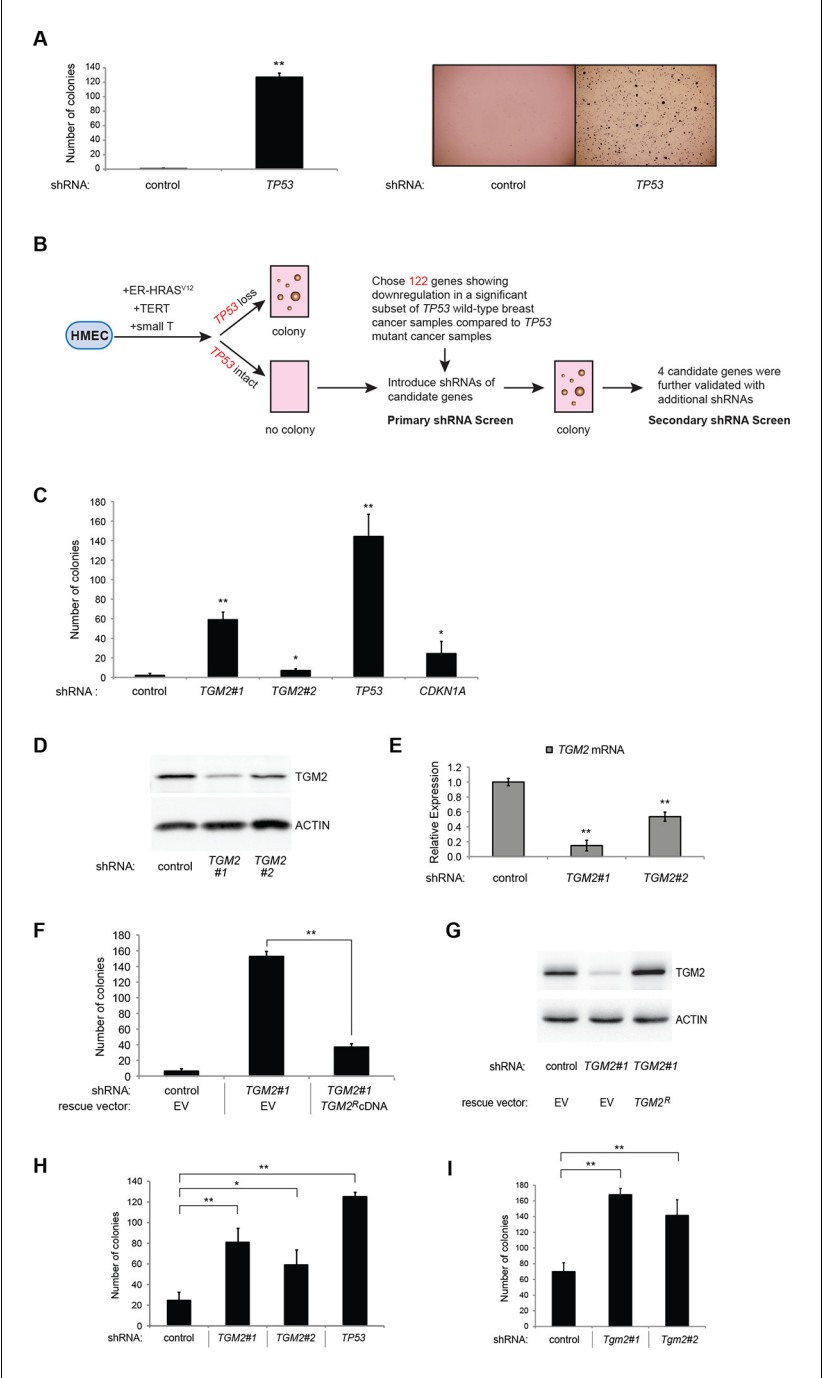

**Figure 1.** TGM2 suppresses transformation of primary human mammary epithelial cells in soft agar. (**A**) HMEC[TERT/ST/ER-RasV12] cells were transduced with retroviral vectors encoding control or *TP53* shRNAs and plated in soft agar in medium with 4-OHT (to activate Ras[V12]). Growth supplements (EGF, insulin, hydrocortisone) were withheld for the first 3 days. Results (left panel) shown are the average colony number ± SD in biological triplicates. Representative MTT-stained colonies are shown in the right panel. (**p<0.01 compared to control cells, student's *t*-test) (**B**) Flow diagram for the shRNA screen. The candidate gene list of 122 genes is selected by comparing genes with lower expression in a significant number of *TP53* wild-type tumor samples versus *TP53* mutant tumor samples using expression array (GSE3494) consisting of 251 breast cancer samples with *TP53* mutation status, followed by designing and cloning of shRNAs against the genes. These genes could be under selective pressure to lose expression only in *TP53* wild-type tumors, thus could be potential members of the TP53 pathway. HMEC[TERT/ST/ER-RasV12] cells were generated by overexpressing TERT, SV40 small T antigen and ER-HRAS[V12]. The shRNAs against the candidate genes were introduced into the cells and observed for colony formation in soft agar in the primary and secondary screen. (**C**) Soft agar analysis for HMEC[TERT/ST/ER-RasV12] cells expressing control, *TGM2*- (denoted as *TGM2*#1 or *TGM2*#2), *TP53*-, or *CDKN1A*- shRNAs using the conditions described in (**B**). Quantification shows average colony number ± SD in biological triplicates. (*p<0.05; **p<0.01 compared to control cells, student's *t*-test) (**D**) Knockdown efficiency of *TGM2* with two independent shRNAs. TGM2 protein expression was

*Figure 1 continued on next page*

*Figure 1 continued*

analyzed by Western blotting. $\beta$-ACTIN serves as the loading control. (**E**) *TGM2* mRNA expression was quantified by qPCR, normalized to *TBP* expression and to control vector in biological triplicates, and represented as the average fold change ± SD. (**\*\*p<0.01 compared to control cells, student's *t*-test) (**F**) Soft agar assay analysis of HMEC<sup>TERT/ST/ER-RasV12</sup> cells transduced with a retrovirus expressing mCherry with either control or *TGM2*#1 shRNAs. The populations were verified to have more than 70% mCherry positive cells, and additionally transduced and selected to express an empty vector (EV) or a shRNA-resistant *TGM2* cDNA (*TGM2*<sup>R</sup>cDNA) by retroviruses. Quantification shows average colony number ± SD in biological triplicates. (**\*\*p<0.01, student's *t*-test) (**G**) Western blot analysis of TGM2 protein expression for (**F**). $\beta$-ACTIN serves as the loading control. (EV, Empty Vector) (**H**) Soft agar assay analysis of BJ<sup>TERT/ST/ER-RasV12/shp16</sup> cells transduced with a retrovirus expressing control, *TGM2*-, or *TP53*- shRNA. Quantification shows average colony number ± SD in biological triplicates. (**\*p<0.05; \*\*p<0.01, compared to control cells, student's *t*-test) (**I**) Soft agar assay analysis of NIH 3T3<sup>ER-RasV12</sup> cells transduced with a retrovirus expressing control or *Tgm2*-shRNAs. Quantification shows average colony number ± SD in biological triplicates. (**\*\*p<0.01 compared to control cells, student's *t*-test).

The following figure supplements are available for figure 1:

**Figure supplement 1.** The effect of TP53 on colony formation in HMEC<sup>TERT/ST/ER-RasV12</sup> cells in the presence of EGF, insulin, and hydrocortisone.

**Figure supplement 2.** Knockdown efficiency of *TP53* shRNA.

**Figure supplement 3.** Knockdown efficiency of *TP53* shRNA for *Figure 1A*.

**Figure supplement 4.** Generating the candidate gene list for screening.

**Figure supplement 5.** Generating the candidate gene list for screening.

**Figure supplement 6.** Generating the candidate gene list for screening.

**Figure supplement 7.** Pictures of soft agar assay for *Figure 1C*.

**Figure supplement 8.** Protein expression for *Figure 1C*.

**Figure supplement 9.** Pictures of soft agar assay for *Figure 1H*.

**Figure supplement 10.** Protein expression for *Figure 1H*.

**Figure supplement 11.** Pictures of soft agar assay for *Figure 1I*.

**Figure supplement 12.** Protein expression for *Figure 1I*.

We further validated the effect of *TGM2* knockdown on colony formation in soft agar using different cell types. Human foreskin fibroblast BJ cells were retrovirally transduced with TERT, ER-HRAS<sup>V12</sup>, SV40 small T, and *p16*<sup>INK4a</sup> shRNA (to disrupt the Rb pathway). Two independent shRNAs against *TGM2* were further transduced into these cells and the number of colonies was evaluated. Consistent with the results from HMECs, knockdown of *TGM2* enhanced the colony formation in BJ<sup>TERT/ST/ER-RasV12/shp16</sup> cells (*Figure 1H* and *Figure 1—figure supplements 9* and *10*). A similar result was also obtained with mouse embryonic fibroblast NIH 3T3 cells expressing ER-HRAS<sup>V12</sup> (*Figure 1I* and *Figure 1—figure supplements 11* and *12*). These results suggest that TGM2 has a tumor suppressive role not only in human mammary epithelial cell (HMECs), but also in BJ human fibroblasts and NIH 3T3 mouse fibroblasts.

### *TGM2* expression is dependent on TP53

TGM2 could act downstream, upstream, or as a co-regulator of TP53 to support tumor suppression. To distinguish between these possibilities, we assessed *TP53* expression and activity in *TGM2* knockdown HMEC<sup>TERT/ST/ER-RasV12</sup> cells. Depletion of *TGM2* expression did not reduce *TP53* expression, or its transcription factor activity, as measured by the expression of TP53 target genes such as *CDKN1A* and *MDM2* (*Figure 2A*). Thus, TGM2 is not an upstream regulator or a co-factor of TP53. In contrast, we observed a significant reduction in *TGM2* mRNA and protein expression in cells expressing *TP53*

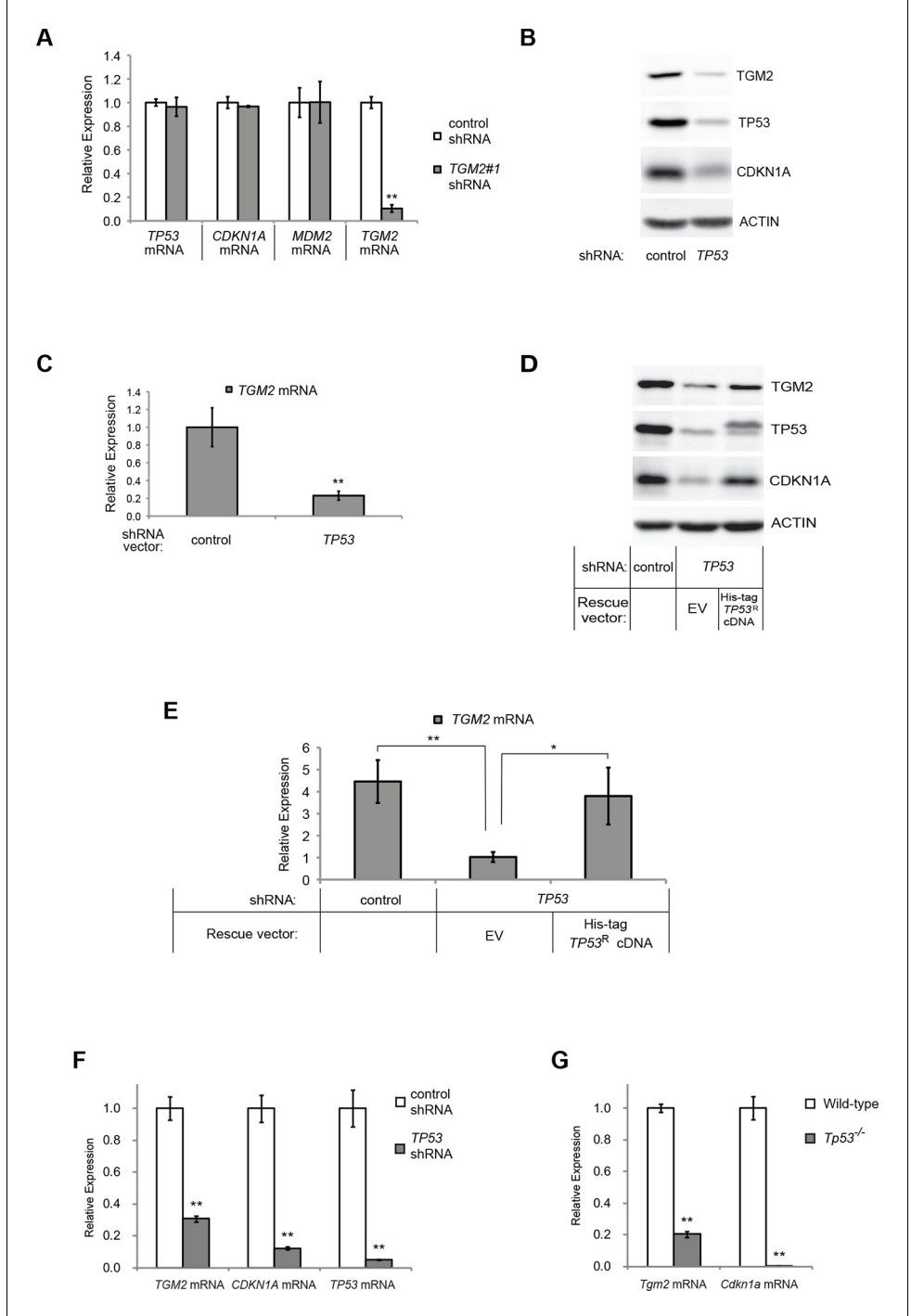

**Figure 2.** *TGM2* expression is dependent on TP53. (**A**) qRT-PCR analysis of the effect of control or *TGM2* shRNAs (denoted respectively as control and *TGM2*#1) on mRNA expression of either *TP53, CDKN1A, MDM2,* or *TGM2* in HMEC[TERT/ST/ER-RasV12] cells. The levels of mRNA were normalized to *TBP* expression and to control cells. The data indicate the average ± SD of biological triplicates. (**p<0.01, student's *t*-test to control cells) (**B**) Western blot analysis of HMEC[TERT/ST/ER-RasV12] cells stably transduced with retroviruses expressing control or *TP53* shRNAs. *β*-ACTIN serves as the loading control. (**C**) qRT-PCR analysis of the cells in (**B**). The levels of *TGM2* mRNA were normalized to *TBP* expression and to control cells. The data indicate the average ± SD of biological triplicates. (**p<0.01, student's *t*-test to control cells) (**D**) Western blot analysis of HMEC[TERT/ST/ER-RasV12] cells stably transduced with a retrovirus expressing mCherry with either control or *TP53* shRNAs. The populations were verified to have more than 70% mCherry positive cells, and then retrovirally-transduced and selected to express an empty vector or a 6x His-tag *TP53* shRNA-resistant overexpression vector (His-tag *TP53*[R]cDNA). *β*-ACTIN serves as the loading control. (EV, Empty Vector) (**E**) qRT-PCR analysis of cells in (**D**) for *TGM2* mRNA normalized to *TBP* expression and to control cells. The data indicate the average ± SD of biological triplicates. (*p<0.05; **p<0.01, student's *t*-test) (**F**) qRT-PCR analysis of

*Figure 2 continued on next page*

*Figure 2 continued*

BJ[TERT/ST/ER-RasV12/shp16] cells stably transduced with retroviruses expressing control or *TP53* shRNAs. The levels of mRNA were normalized to *TBP* expression and to control cells. The data indicate the average ± SD of biological triplicates. (**p<0.01, student's *t*-test to control cells) (**G**) qRT-PCR analysis of wild-type and *Tp53*[-/-] MEFs. The levels of mRNA were normalized to *Gapdh* expression and to control cells. The data indicate the average ± SD of biological triplicates. (**p<0.01, student's *t*-test to control cells).

shRNAs (*Figure 2B and C*), suggesting that *TGM2* is induced by TP53. Conversely, reconstitution of *TP53* expression in HMEC[TERT/ST/ER-RasV12] cells co-expressing a *TP53* shRNA with a *TP53* shRNA-resistant cDNA (His-tag TP53[R]) restored TGM2 protein and mRNA expression (*Figure 2D and E*), further validating that *TGM2* is regulated by TP53. Consistent with these findings, the levels of *TGM2* mRNA was lower in BJ[TERT/ST/ER-RasV12/shp16] cells expressing *TP53* shRNA compared to control BJ[TERT/ST/ER-RasV12/shp16] cells (*Figure 2F*). Furthermore, a reduction in *TGM2* mRNA was also observed in *Tp53* knockout MEFs compared to wild-type MEFs (*Figure 2G*). Together, these data strongly suggest that *TGM2* is regulated by TP53.

## *TGM2* is a potential transcriptional target gene of TP53

To examine if *TGM2* is induced by TP53 activation, we treated HMEC[TERT/ST/ER-RasV12] cells and BJ[TERT/ST/ER-RasV12/shp16] with Nutlin-3a, a small molecule inhibitor of the TP53-MDM2 interaction (*Tovar et al., 2006*; *Vassilev et al., 2004*). MDM2 promotes degradation of TP53, thus inhibiting this interaction with Nutlin-3a effectively stabilizes TP53. In both cell types, *TGM2* expression was increased by Nutlin-3a treatment in a TP53-dependent manner (*Figure 3A and B*), suggesting that *TGM2* could be a transcriptional target gene of TP53.

It has been reported that the *TGM2* promoter contains two predicted binding sites for TP53, although these sites were not tested for TP53 binding and TP53-mediated transactivation (*Ai et al., 2012*). Several other potential TP53 binding motifs within the *TGM2* promoter were also predicted using computer software, p53MH algorithm described previously (*Hoh et al., 2002*). To determine if the *TGM2* promoter contains TP53 target sites, we engineered a reporter system which contained a luciferase gene under the control of the partial *TGM2* promoter sequences (*Figure 3C*). We investigated non-overlapping regions from -5980 to -78 base pairs upstream of the *TGM2* translational start site (ATG). Co-transfection of the -1530 to -78 region of the *TGM2* promoter/luciferase reporter with a TP53 expression vector into TP53-null H1299 cells significantly increased luciferase activity (*Figure 3C*). We evaluated several deletion constructs corresponding to this region, and unexpectedly found that an element from -159 to -78 of the *TGM2* promoter, which does not contain a TP53 binding consensus, was necessary and sufficient for TP53-mediated transactivation (*Figure 3D*). Further deletion of 5 or 10 nucleotides at the 5'- or 3'-end from this 82 bp element significantly reduced luciferase activity (*Figure 3—figure supplement 1*), suggesting that it represents the minimal region for TP53-mediated activation within the *TGM2* promoter (*Figure 3—figure supplement 2*). Further, we used a TP53 antibody for chromatin immunoprecipitation (ChIP) and found that this genomic region of the *TGM2* promoter was specifically immunoprecipitated with endogenous TP53 from HMEC[TERT/ST/ER-RasV12] cells (*Figure 3E* and *Figure 3—figure supplement 2*), suggesting that TP53 directly binds to this element to regulate *TGM2* expression. Together, these findings suggest that the *TGM2* promoter contains a novel target site for TP53 binding and activation.

## Depletion of growth supplements induces TP53-dependent autophagy

TP53 is known to induce autophagy, the catabolic breakdown of cellular components by the lysosome (*Budanov and Karin, 2008*; *Crighton et al., 2006*; *Feng et al., 2005*; *Kenzelmann Broz et al., 2013*), and autophagy can have a tumor suppressive function (*Karantza-Wadsworth et al., 2007*; *Mathew et al., 2007*). Various studies have shown that the absence of growth factor signaling can also induce autophagy (*Cheng et al., 2010*; *Eom et al., 2014*; *Lum et al., 2005*). We wanted to determine whether the deprivation of growth supplements (EGF, insulin, and hydrocortisone) in our soft agar assay induced autophagy in HMEC[TERT/ST/ER-RasV12] cells and, if so, whether autophagy was TP53-dependent. We used a well-established approach employing a tandem RFP-GFP-LC3 fusion construct to monitor autophagy by fluorescence microscopy (*Mizushima et al., 2010*). MAP1LC3A (known as LC3) is modified to LC3-II, a critical component of autophagosomes, which engulf cellular

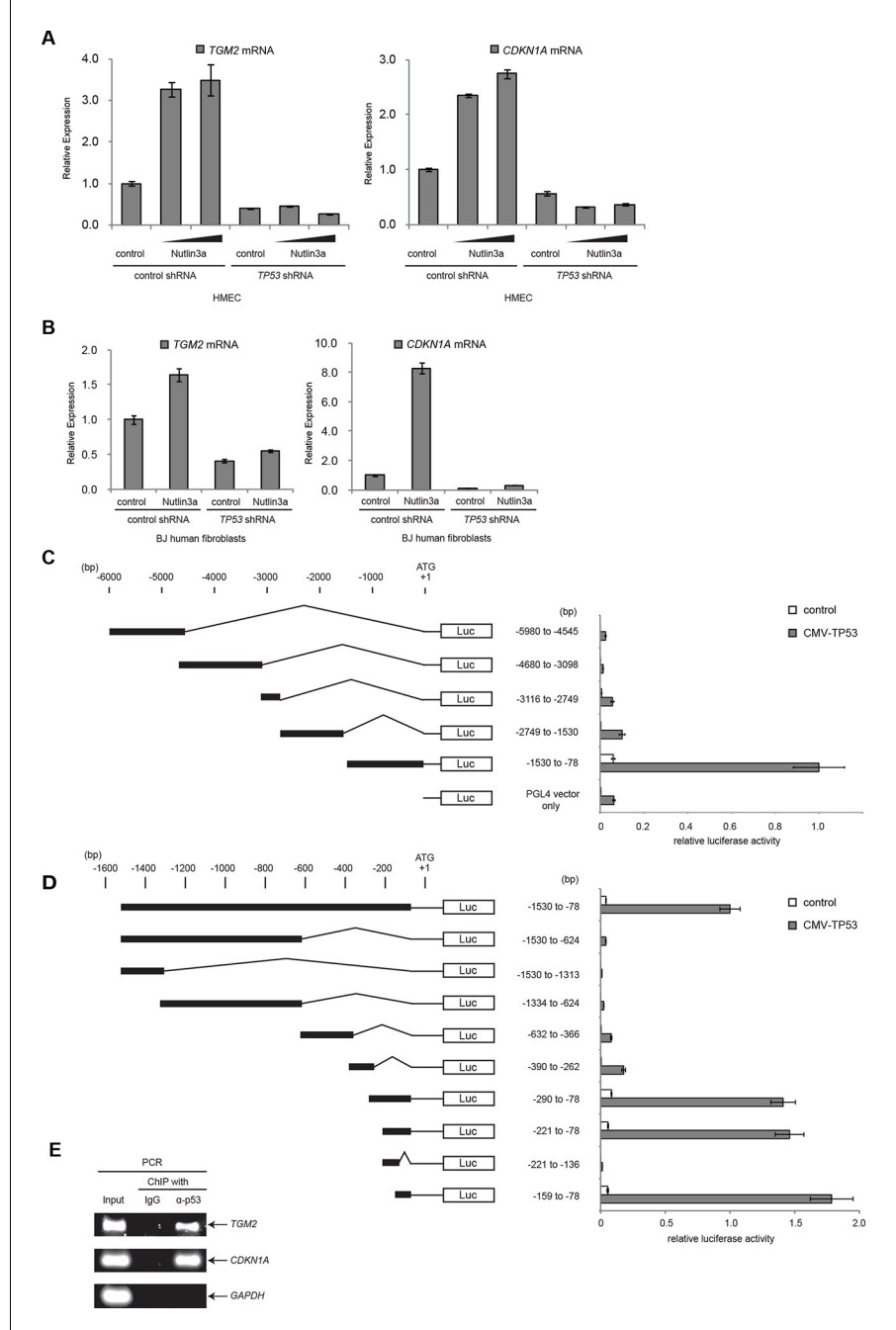

**Figure 3.** *TGM2* is a potential target gene of TP53. (**A**) qRT-PCR analysis of HMEC[TERT/ST/ER-RasV12] cells stably transduced with retroviruses expressing control or *TP53* shRNAs. Cells were treated with Nutlin-3a (5 μM or 10 μM) for 2 days. The levels of mRNA were normalized to *TBP* expression and to control cells. *CDKN1A* is used as the positive control to see TP53 activation. The data indicate the average ± SD of biological triplicates. (**B**) qRT-PCR analysis of BJ[TERT/ST/ER-RasV12/shp16] cells stably transduced with retroviruses expressing control or *TP53* shRNAs. Cells were treated with Nutlin-3a (10 μM) for 2 days. The levels of mRNA were normalized to *TBP* expression and to control cells. *CDKN1A* is used as the positive control to see TP53 activation. The data indicate the average ± SD of biological triplicates. (**C** and **D**) Luciferase reporter assays using a series of promoter deletion mutants of the *TGM2* gene. The number (bp) indicates the position relative to the translational start site (ATG). Reporter plasmids containing the indicated deletion constructs were transfected into H1299 cells with control or TP53 plasmid, and luciferase activity was monitored. The average value of the luciferase activity from the cells transfected with CMV-TP53 and the reporter plasmid containing *TGM2* (-1530 to -78) promoter fragment is set at 1, and the relative activity is shown. The data indicate the average ± SD of biological triplicates. (**E**) TP53 binds to the *TGM2*

*Figure 3 continued on next page*

*Figure 3 continued*

promoter. ChIP assay was performed with an antibody detecting endogenous TP53, or IgG (negative control) using HMEC$^{TERT/ST/ER-RasV12}$ cells. The potential TP53 response element in the *TGM2* promoter identified in (D) was analyzed by PCR. *CDKN1A* and *GAPDH* are served as the positive and negative control respectively.

The following figure supplements are available for figure 3:

**Figure supplement 1.** Luciferase reporter assays of *TGM2* promoter.

**Figure supplement 2.** Diagram of the *TGM2* promoter region.

components and fuse with lysosomes during autophagy (*Kimmelman, 2011*). In the absence of autophagy, cells display diffuse colocalization of both red and green signals from RFP-GFP-LC3. In contrast, the fusion of autophagosomes with lysosomes during autophagy results in rapid quenching of the GFP signal from RFP-GFP-LC3, since it is more sensitive to the acidic conditions of the autolysosomal lumen than RFP. Therefore, RFP signals without GFP, visualized as red punctae in the cytoplasm, represent acidic compartments such as autolysosomes and signify autophagic flux (*Kimura et al., 2007*; *Klionsky et al., 2012*; *Mizushima, 2009*; *Wu et al., 2010*).

Since we used a TERT-H2B-GFP construct, which expresses TERT and H2B-GFP, to generate HMEC$^{TERT/ST/ER-RasV12}$ cells (*Kolfschoten et al., 2005*), there is a ubiquitious GFP signal in the nucleoplasm due to the nuclear localization signal of histone H2B. Thus, the green signal from H2B-GFP overlapped with the green signal from RFP-GFP-LC3 in nuclei. Nevertheless, we were still able to monitor autophagic flux by the presence of red punctae in the cytoplasm.

About 80% of HMEC$^{TERT/ST/ER-RasV12}$ cells (denoted as *TP53$^{+/+}$*) transiently transfected with the tandem RFP-GFP-LC3 construct and cultured in the absence of growth supplements displayed extensive red punctae and no green signal in the cytoplasm, indicative of autophagy (*Figure 4A and B*) (*Boland et al., 2008*). To determine whether the observed autophagy was dependent on *TP53*, we generated *TP53* knockout HMEC$^{TERT/ST/ER-RasV12}$ cells (denoted as *TP53$^{-/-}$*) by CRISPR/Cas technology. After transfection of CRISPR plasmids, single clones were isolated and mutation of the *TP53* locus was validated by DNA sequencing (*Figure 4—figure supplement 1*). In contrast to *TP53$^{+/+}$* cells, only about 40% of *TP53$^{-/-}$* cells transfected with RFP-GFP-LC3 exhibited red punctae, and more cells showed complete overlap of green and red fluorescence signals (*Figure 4A and B*). To exclude potential effects from the clonal selection of *TP53$^{-/-}$* cells, we also transfected the RFP-GFP-LC3 construct into cells expressing *TP53* shRNA. Consistent with the results from *TP53$^{-/-}$* cells, *TP53* knockdown cells displayed a lower percentage of cells having red punctae without green signal in the cytoplasm compared to control cells (*Figure 4C and D*). These observations suggest that the depletion of growth supplements induces TP53-dependent autophagy, which may limit colony formation in our soft agar assay.

To further assess the role of TP53 in autophagic flux, we treated cells with chloroquine (CQ), an agent that prevents the acidification of lysosomes and inhibits autophagic flux by preventing lysosomal protein degradation (*Shintani and Klionsky, 2004*). During active autophagic flux, LC3-I is conjugated to phosphatidylethanolamine (PE) to form LC3-II, a widely-used autophagic marker (*Kabeya et al., 2004*). The addition of chloroquine to cells undergoing autophagy prevents LC3-II degradation by lysosomal enzymes, leading to the accumulation of LC3-II protein and an increase in the ratio of LC3-II to LC3-I (*Klionsky et al., 2012*; *Mizushima and Yoshimori, 2007*). However, if cells do not have active autophagic flux, the levels of both LC3-I and LC3-II will be limited and their ratio will not be affected by chloroquine treatment (*Mizushima and Yoshimori, 2007*). The addition of chloroquine to parental HMEC$^{TERT/ST/ER-RasV12}$ cells and to HMEC$^{TERT/ST/ER-RasV12}$ cells expressing control shRNAs triggered an increase in the LC3-II to LC3-I ratio, indicating rapid autophagic flux (*Figure 4E*, lane 1, 2, 7, and 8) (*Klionsky et al., 2012*). However, *TP53* knockdown or knockout cells displayed decreased LC3-I and LC3-II protein levels, and minimal accumulation of LC3-II by chloroquine treatment, indicating that TP53 contributes to active autophagic flux (*Figure 4E*).

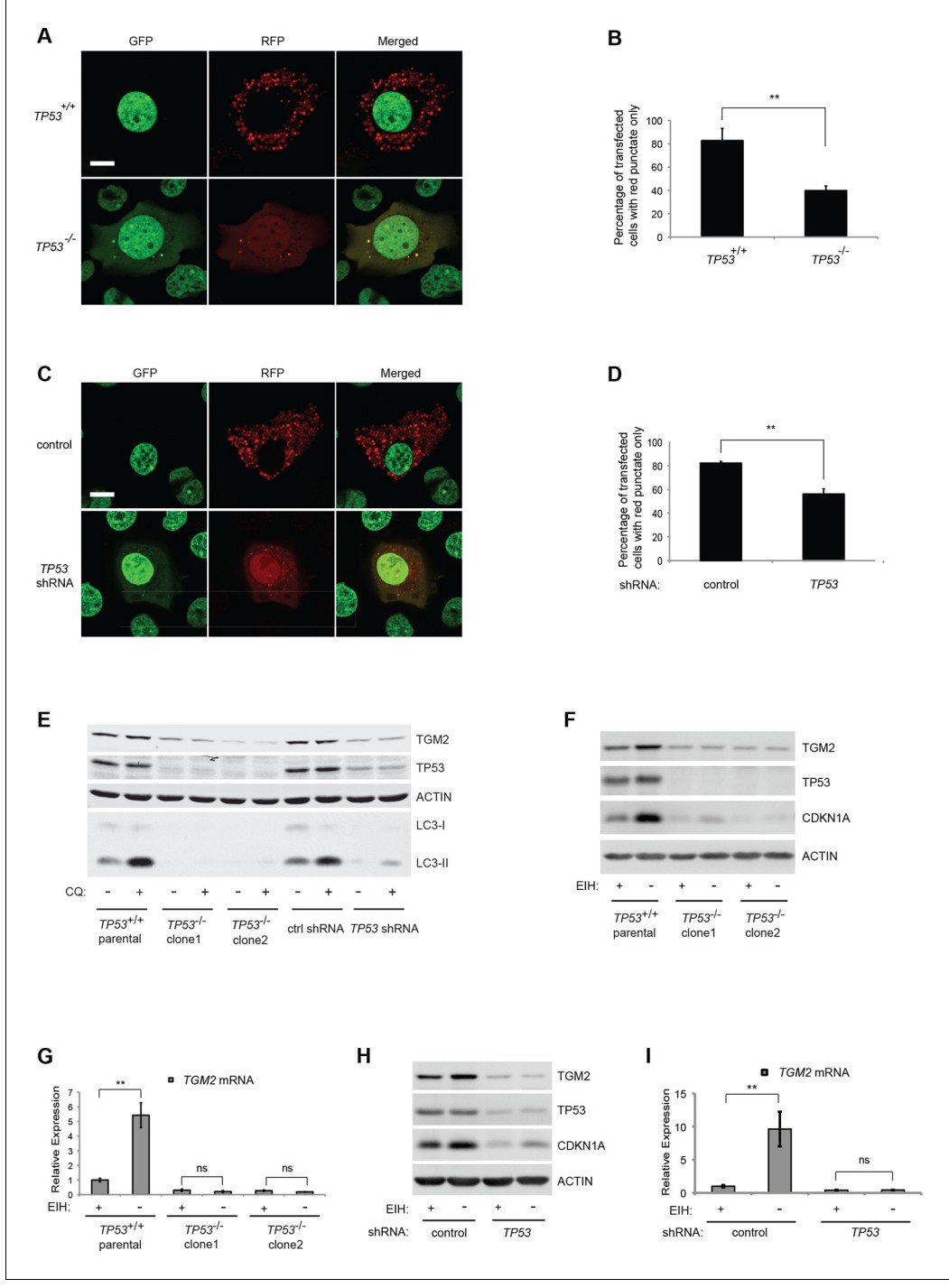

**Figure 4.** Absence of growth supplements induces TP53-dependent autophagy. (**A**) Formation of red punctae following autophagy induction in the absence of growth supplements. HMEC[TERT/ST/ER-RasV12] cells (*TP53*[+/+]) or *TP53* CRISPR knockout HMEC[TERT/ST/ER-RasV12] cells (*TP53*[-/-]) cells were transfected with the plasmid RFP-GFP-LC3. Cells were incubated in medium without EGF, insulin, and hydrocortisone for 24 hr before visualization on a confocal microscope. Scale bar: 5 μm (**B**) Quantification of the fraction of red punctate cells within the total population of transfected cells shown in (**A**). Red punctate cells were counted as cells containing only RFP signal without visible overlap of GFP signal in the cytoplasm; transfected cells were counted as cells containing either RFP signals or a mix of RFP and GFP signals in the cytoplasm (>250 cells were counted). (**p<0.01, student's *t*-test) (**C**) Control or *TP53* knockdown (*TP53* shRNA) HMEC[TERT/ST/ER-RasV12] cells were seeded, treated and visualized as in (**A**). (**D**) Quantification of the fraction of red punctate cells within the total population of transfected cells

*Figure 4 continued*

shown in (**C**), treated as in (**B**). (\*\*p<0.01, student's *t*-test) (**E**) Western blot analysis of *TP53*⁺/⁺ cells, *TP53*⁻/⁻ CRISPR knockout HMEC^TERT/ST/ER-RasV12 clones, as well as control and *TP53* shRNA cells treated with or without chloroquine (CQ, 50 μM, 2 hr) after incubation in medium without EGF, insulin, and hydrocortisone for 24 hr. *β*-ACTIN serves as the loading control. (**F**) Western blot analysis of HMEC^TERT/ST/ER-RasV12 cells and two independent *TP53* CRISPR knockout HMEC^TERT/ST/ER-RasV12 clones. Cells were incubated in medium in the presence or absence of EGF, insulin, and hydrocortisone (denoted as EIH) for 48 hr. *β*-ACTIN serves as the loading control. (**G**) qRT-PCR analysis of cells in (**F**). *TGM2* mRNA expression was normalized to *TBP* mRNA expression. The mean value of *TGM2* mRNA expression in *TP53*⁺/⁺ cells with presence of EIH is set at 1, and relative expression is shown. (\*\*p<0.01, ns: not significant, student's *t*-test) (**H**) Western blot analysis of HMEC^TERT/ST/ER-RasV12 cells expressing either control or *TP53* shRNA. Cells were treated the same as in (**F**). (**I**) qRT-PCR analysis of cells in (**H**). Data are shown as in (**G**).

The following figure supplement is available for figure 4:

**Figure supplement 1.** DNA sequencing of *TP53*⁻/⁻ clones.

## *TGM2* is induced by the depletion of growth supplements in a TP53-dependent manner

To determine if TGM2 is also involved in the autophagy induced by a depletion of growth supplements, we analyzed the expression of TGM2 in the presence or absence of EGF, insulin, and hydrocortisone. Indeed, depletion of these growth supplements increased TGM2 protein and mRNA levels in *TP53* wild-type (*TP53*⁺/⁺) HMEC^TERT/ST/ER-RasV12 cells (*Figure 4F and G*). In contrast, removal of growth supplements in *TP53* knockout HMEC^TERT/ST/ER-RasV12 cells (*TP53*⁻/⁻ clone 1 and clone 2), as well as in *TP53* shRNA cells did not induce TGM2 protein and mRNA expression (*Figure 4F, G, H, and I*). Thus, *TGM2* is induced by the depletion of growth supplements in a TP53-dependent manner, suggesting a potential role for TGM2 in mediating the TP53-induced autophagic program.

## TGM2 promotes autophagic protein degradation and autolysosome clearance at late stages of autophagy

Previous studies have reported that TGM2 promotes autophagy (*D'Eletto et al., 2009*). Therefore, we hypothesized that a depletion of growth supplements induces TP53-dependent autophagy in part through TGM2. To test this directly, we transfected the RFP-GFP-LC3 plasmid into HMEC^TERT/ST/ER-RasV12 cells expressing *TGM2* shRNA and cultured them without growth supplements. Contrary to our expectation, knockdown of *TGM2* did not affect the fraction of cells having red punctae without green signal, unlike knockdown of *TP53* (*Figure 5A and B*). This finding indicates that reduced *TGM2* expression does not prevent fusion of autophagosomes with lysosomes to form autolysosomes. Interestingly, however, we noticed that *TGM2* knockdown cells displayed a two-fold enlargement in the size of red punctae compared to control cells (*Figure 5A and C*). Furthermore, we observed similarly enlarged yellow punctae in both control and *TGM2* knockdown cells after chloroquine treatment (*Figure 5A and C*). Addition of chloroquine blocks the acidification of lysosomes, thereby preventing autophagic protein degradation, enlarging the size of autolysosome and preventing the quenching of GFP fluorescence in the autolysosome. The formation of enlarged red punctae upon knockdown of *TGM2*, even without chloroquine treatment, suggests a defect in the later steps of autophagy such as autolysosome clearance, leading to autolysosome enlargement (*Boland et al., 2008*; *Nixon et al., 2005*).

To test whether TGM2 promotes autophagic protein degradation and autolysosome clearance, we treated control and *TGM2* knockdown cells with chloroquine (CQ) and monitored LC3-I and LC3-II protein levels. A block in autophagic protein degradation and autolysosome clearance in cells would be predicted to lead to increased LC3-II protein levels even in the absence of chloroquine, and the addition of chloroquine in these cells would result in only a minimal increase in LC3-II and in the ratio of LC3-II to LC3-I (*Mizushima and Yoshimori, 2007*). The ratio of LC3-II/LC3-I in control HMEC^TERT/ST/ER-RasV12 cells increased from 1.3-fold to 5.9-fold after addition of chloroquine, indicating rapid autophagic flux in untreated control cells (*Figure 5D*, compare lane 1 and 2) (*Klionsky et al., 2012*). In contrast, knockdown of *TGM2* with independent hairpins (*TGM2#1* and

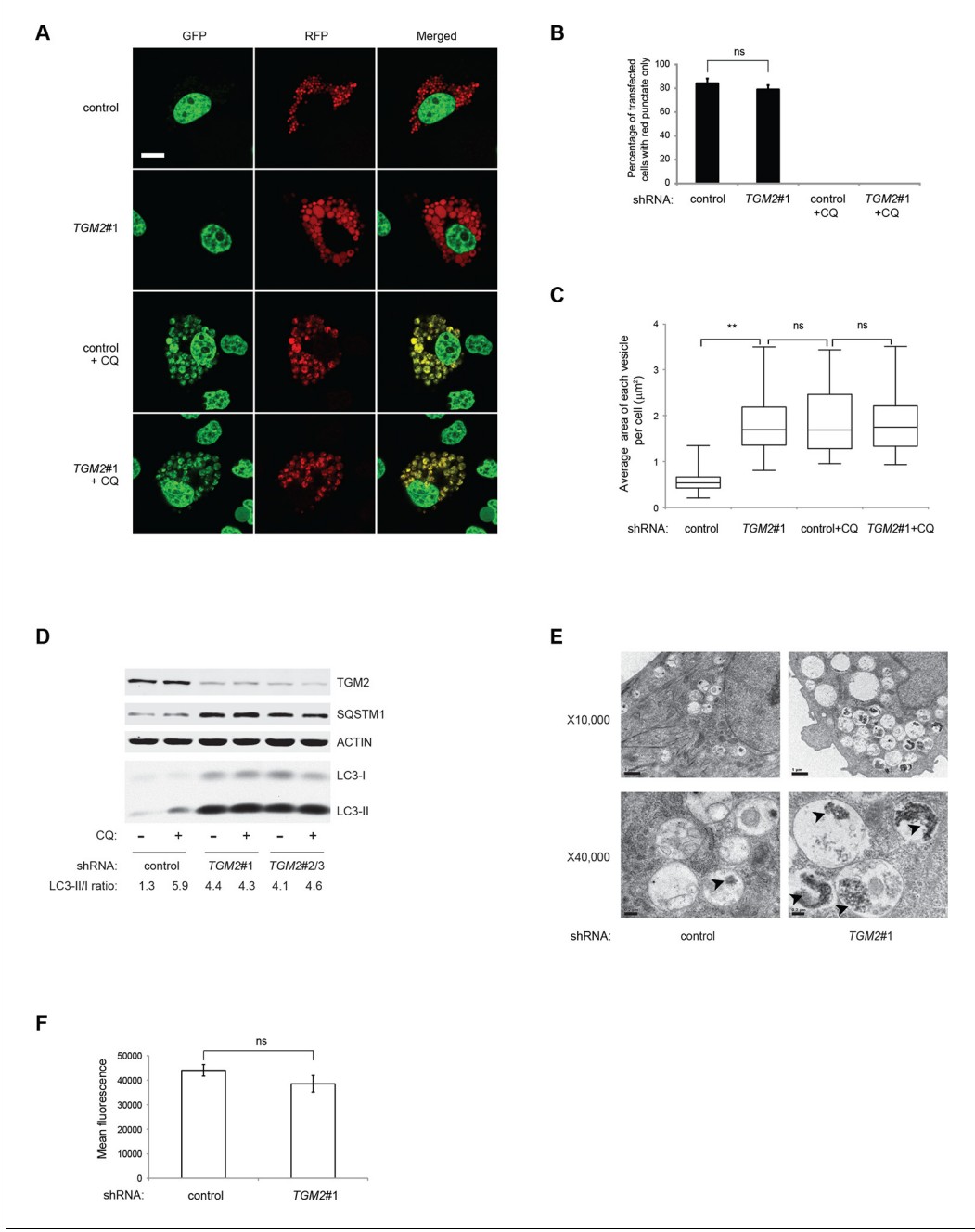

**Figure 5.** Loss of TGM2 expression inhibits autophagic protein degradation and autolysosome clearance. (**A**) Formation of red punctae following autophagy induction in the absence of growth supplements. HMEC[TERT/ST/ER-RasV12] cells (control) and *TGM2* shRNA cells (*TGM2*#1) were transfected with the plasmid RFP-GFP-LC3. Cells were incubated in medium without EGF, insulin, and hydrocortisone with or without chloroquine (20 μM) for 24 hr before visualization on confocal microscope. Scale bar: 5 μm (**B**) Quantification of fraction of cells showing only red punctate within the total population of transfected cells in (**A**). Cells were categorized and counted as in *Figure 4B* (>250 cells counted). (ns: not significant, student's *t*-test) (**C**) Quantification of punctate size in control and *TGM2* shRNA (*TGM2*#1) cells with and without chloroquine treatment. Average area of each vesicle per cell in (**A**) was analyzed by Image J software and represented as box plot. (**p<0.01, ns: not significant, student's *t*-test) (**D**) Western blot analysis of HMEC[TERT/ST/ER-RasV12] cells expressing control and *TGM2* shRNA using independent hairpins (*TGM2*#1 and double hairpins *TGM2*#2/3) treated with or without chloroquine (CQ, 50 μM, 2 hr) after incubation in medium without EGF, insulin, and hydrocortisone for 24 hr. Note that for *TGM2*#2/3, cells were generated by transducing with retrovirus carrying *TGM2*#2 and *TGM2*#3 shRNAs to achieve similar knock-down

*Figure 5 continued on next page*

*Figure 5 continued*

efficiency to *TGM2* shRNA#1. *β*-ACTIN serves as the loading control. (**E**) Transmission electron microscopy images for control and *TGM2* shRNA (*TGM2*#1) cells. Cells were incubated without EGF, Insulin, and hydrocortisone for 24 hr before fixing and imaging. The higher magnification micrograph (x40,000) shows presence of undigested protein aggregates (arrowheads) in autophagic vesicles. Lower magnification was set at x10,000. Scale bar: 1 μm at x10,000 and 0.2 μm at x40,000. (**F**) Flow cytometry analysis of cells stained with lysotracker after incubation in media without EGF, insulin, and hydrocortisone for 24 hr. The mean fluorescence intensity of control and *TGM2* shRNA (*TGM2*#1) cells was quantified by Flowjo software. The data indicate the mean ± SD of biological triplicates. (ns: not significant, student's *t*-test).

*TGM2#2/3*) increased the LC3-II/LC3-I ratio even in the absence of chloroqiune (*Figure 5D*, compare lane 1, 3 and 5), to a similar extent as that observed in control cells after chloroquine treatment (*Figure 5D*, lane 2). Furthermore, the addition of chloroquine did not further elevate the levels of LC3-II or the LC3-II/LC3-I ratio in *TGM2* knockdown cells compared to control cells (*Figure 5D*, lane 3–6), suggesting that *TGM2* knockdown leads to a defect in autophagic protein degradation.

The SQSTM1/p62 protein is known to bind ubiquitinated proteins and transport them to the autophagy machinery for their degradation (*Kimmelman, 2011*). A block in autophagy leads to the accumulation of SQSTM1, since the SQSTM1 protein itself is degraded by autophagy (*Mizushima and Yoshimori, 2007*). We observed an accumulation of SQSTM1 protein in *TGM2* knockdown cells (*Figure 5D*), providing additional evidence that loss of *TGM2* impairs autophagic protein degradation and autolysosome clearance.

To further clarify how loss of *TGM2* impacts autophagy, we used transmission electron microscopy to visualize autophagic vesicles at the ultrastructural level (*Mizushima et al., 2010*). We observed an increase in the size and number of vesicles, as well as an accumulation of undigested protein, seen as black aggregates within the vesicles in *TGM2* knockdown cells compared to control cells, consistent with impairment in autophagic protein degradation and autolysosome clearance (*Figure 5E*).

To determine if loss of *TGM2* altered the acidity of autolysosomes, thereby preventing protein degradation, we incubated cells with Lysotracker Red, a fluorescent dye which labels acidic vesicles in living cells. We used flow cytometry to quantify the fluorescence intensity, and thus the acidity of vesicles (*Chikte et al., 2014*). Interestingly, we did not observe a significant difference in the fluorescence levels between control and *TGM2* knockdown cells (*Figure 5F*), consistent with our observation that the percentage of cells having red punctae without green signal was similar between control and *TGM2* knockdown cells transfected with the RFP-GFP-LC3 construct (*Figure 5A and B*). These data suggest that the loss of *TGM2* does not alter the acidity of autolysosomes, but rather impairs their content degradation and clearance. Taken together, our results suggest a role of TGM2 in autophagy by promoting autophagic protein degradation and autolysosome clearance.

## CDKN1A and TGM2 play distinct tumor suppressive roles in the TP53 pathway

We showed that loss of either *TGM2* or *CDKN1A* could stimulate colony formation in soft agar (*Figure 1C*). The major function of CDKN1A is to promote cell cycle arrest (*Abbas and Dutta, 2009*; *Chang et al., 2000*), whereas we show here that one function of TGM2 is to promote autophagy by facilitating autophagic protein degradation and autolysosome clearance (*Figure 5*). Next, we examined whether these different functions of TGM2 and CDKN1A cooperate to protect against cell transformation. To this end, we performed combined knockdown of *TGM2* and *CDKN1A* in HMEC-TERT/ST/ER-RasV12 cells to determine the effect on colony formation (*Figure 6A* and *Figure 6—figure supplement 1*). We used two independent shRNAs against *TGM2* (#1 and #2). *TGM2*#1 shRNA showed a higher knockdown efficiency compared to *TGM2*#2 shRNA (*Figure 1D and E*, and *Figure 6—figure supplement 2*). We found that the number of colonies formed with the simultaneous knockdown of *TGM2* and *CDKN1A* was significantly greater than with the knockdown of each gene individually (*Figure 6A*, column 2, 3, 4, 8, and 10 and *Figure 6—figure supplement 1*). Unexpectedly, double knockdown of *CDKN1A/TGM2*#1 gave rise to more colonies than single knockdown of *TP53* (*Figure 6A*, column 5 and 8). This could be due to the residual expression of TGM2 in *TP53* knockdown cells compared to *CDKN1A/TGM2*#1 knockdown cells, arising from the insufficient

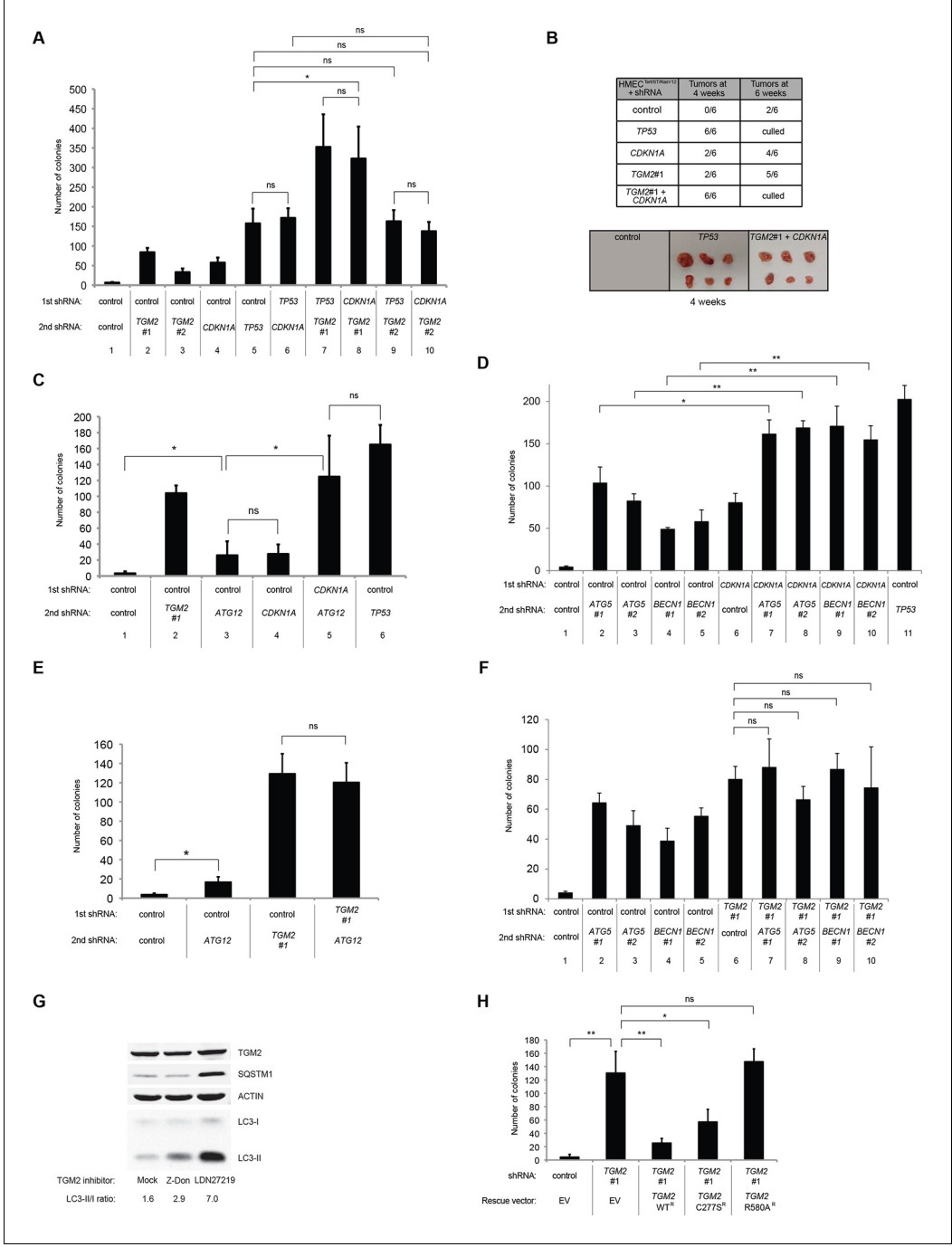

**Figure 6.** Loss of TGM2 expression synergizes with loss of CDKN1A to promote the transformation of HMEC[TERT/ST/ER-RasV12] cells. (**A**) Soft agar assay analysis of HMEC[TERT/ST/ER-RasV12] cells transduced with a retrovirus expressing mCherry with either control, *CDKN1A* or *TP53* shRNAs. The populations were verified to have more than 70% mCherry positive cells. The cells were then additionally transduced with the indicated shRNA constructs, selected with 4 μg/ml of blasticidin, and evaluated by soft agar assay analysis. The results shown are the average colony number ± SD from biological triplicates. (*p<0.05, ns: not significant, student's *t*-test) (**B**) Formation of tumors in NOD/SCID mice. 500,000 HMEC[TERT/ST/RasV12] cells transduced with the indicated vectors were injected subcutaneously in mice (n=6). (Top) The number of tumors observed after 4- and 6-weeks from the time of injection. (Bottom) Pictures of tumors excised 4 weeks after injection. (**C** and **D**) HMEC[TERT/ST/ER-RasV12] cells were transduced with a retrovirus expressing mCherry with either control or *CDKN1A* shRNAs. The populations were verified to have more than 70% mCherry positive cells. The cells were then additionally transduced with the indicated shRNA constructs, selected with 4 μg/ml of blasticidin, and evaluated by soft agar assay analysis. The results shown are the average colony number ± SD from biological triplicates. (*p<0.05, **p<0.01, ns: not significant, student's *t*-test) (**E** and **F**) HMEC[TERT/ST/ER-RasV12] cells were transduced with a retrovirus expressing mCherry with either control or *TGM2 (TGM2*#1) shRNAs. The populations were verified to have more than 70% mCherry positive cells. The cells were then additionally transduced with the indicated shRNA constructs, selected with 4 μg/ml of blasticidin, and
*Figure 6 continued on next page*

*Figure 6 continued*

evaluated by soft agar assay analysis. The results shown are the average colony number ± SD from biological triplicates. (*p<0.05, ns: not significant, student's *t*-test). (**G**) Western blot analysis of HMEC^TERT/ST/ER-RasV12^ cells treated with TGM2 inhibitors. The cells were incubated in medium without EGF, insulin, and hydrocortisone in the presence of Z-DON (50 µM) or LDN 27219 (10 µM) for 24 hr. β-ACTIN serves as the loading control. (**H**) Soft agar assay analysis of HMEC^TERT/ST/ER-RasV12^ cells transduced with a retrovirus expressing mCherry with either control or *TGM2* (*TGM2*#1) shRNAs. The populations were verified to have more than 70% mCherry positive cells, and additionally transduced and selected with 4 µg/ml of blasticidin to express an empty vector (EV) or a *TGM2* cDNA resistant to *TGM2* shRNA wild-type (WT), C277S, or R580A mutants (denoted as *TGM2* WT^R^, *TGM2* C277S^R^, or *TGM2* R580A^R^ cDNA) by retroviruses. Quantification shows average colony number ± SD in biological triplicates. (**p<0.01, *p<0.05, ns: not significant, student's *t*-test).

The following figure supplements are available for figure 6:

**Figure supplement 1.** Pictures of soft agar assay for *Figure 6A*.

**Figure supplement 2.** Protein expression for *Figure 6A*.

**Figure supplement 3.** Pictures of soft agar assay for *Figure 6C*.

**Figure supplement 4.** Protein expression for *Figure 6C*.

**Figure supplement 5.** Protein expression of BECN1, ATG5, ATG12, and LC-3.

**Figure supplement 6.** Pictures of soft agar assay for *Figure 6D*.

**Figure supplement 7.** Protein expression for *Figure 6D*.

**Figure supplement 8.** Pictures of soft agar assay for *Figure 6E*.

**Figure supplement 9.** Protein expression for *Figure 6E*.

**Figure supplement 10.** Pictures of soft agar assay for *Figure 6F*.

**Figure supplement 11.** Protein expression for *Figure 6F*.

**Figure supplement 12.** Pictures of soft agar assay for *Figure 6H*.

**Figure supplement 13.** Protein expression for *Figure 6H*.

down regulation of TGM2 by a single *TP53* shRNA (compare lane 1, 5, and 8 in *Figure 6—figure supplement 2*). Indeed, cells expressing both *TGM2*#2 shRNA, which has a lower knockdown efficiency of *TGM2* (*Figure 1D and E*, and *Figure 6—figure supplement 2*), and *CDKN1A* shRNA generated a similar number of colonies as *TP53* knockdown cells (*Figure 6A*, column 5 and 10). Knockdown of *CDKN1A* and *TP53* together generated a similar number of colonies as loss of *TP53* alone, indicating that these two genes act in the same pathway (*Figure 6A*, column 5 and 6). Furthermore, the effect of *TP53/TGM2*#2 knockdown was also comparable to the potency of *TP53* knockdown alone for inducing colony formation in soft agar (*Figure 6A*, column 5 and 9). *TP53/CDKN1A* knockdown and *CDKN1A/TGM2*#2 knockdown cells also did not generate significantly different colony numbers (*Figure 6A*, column 6 and 10). These data suggest that CDKN1A and TGM2 suppress colony formation mainly through the TP53 pathway. The cooperative effect of *CDKN1A* and *TGM2* knockdown indicates that they provide complementary contributions to tumor suppression and that loss of each gene function is critical for oncogenic transformation.

Next, we investigated whether combined loss of *TGM2* and *CDKN1A* would enhance tumorigenesis of HMEC^TERT/ST^ cells expressing constitutively active HRAS^V12^ (denoted as HMEC^TERT/ST/RasV12^ cells) in a xenograft model. Subcutaneous injection of HMEC^TERT/ST/RasV12^ cells expressing control shRNAs into immunocompromised NOD/SCID mice did not lead to tumors after 4 weeks, and formed only a few, small tumors after 6 weeks (*Figure 6B*), which is consistent with the small number

of colonies observed in soft agar (*Figure 6A*). In contrast, HMEC$^{TERT/ST/RasV12}$ cells in which *TP53* expression was reduced, and more importantly in which both *TGM2* and *CDKN1A* expression were simultaneously reduced, formed six tumors out of six subcutaneous injections in NOD/SCID mice after 4 weeks (*Figure 6B*). HMEC$^{TERT/ST/RasV12}$ cells expressing shRNAs targeting either *TGM2* or *CDKN1A* alone produced tumors with a reduced penetrance and increased latency compared to those expressing *TP53* shRNA (two tumors out of six injections after 4 weeks) (*Figure 6B*), consistent with their performance in the soft agar assay (*Figure 6A*). Taken together, our data show that loss of both *TGM2* and *CDKN1A* expression allows HMECs to overcome TP53-dependent tumor suppression in vitro and in vivo.

## Autophagy and cell cycle arrest are both required for complete TP53-dependent tumor suppression

We hypothesized that reduced *TGM2* expression enabled colony formation in soft agar by interfering with TP53-dependent autophagy. Therefore, inhibiting autophagy by other means should phenocopy *TGM2* knockdown and also synergize with *CDKN1A* knockdown to promote colony formation in soft agar as shown in *Figure 6A*. To test this hypothesis, we evaluated the role of *ATG12*, which controls autophagosome formation (*Kimmelman, 2011*), in tumor suppression. We expressed *ATG12* shRNA (*Lock et al., 2011*) in HMEC$^{TERT/ST/ER-RasV12}$ stably expressing control- or *CDKN1A*-shRNAs and analyzed colony formation in soft agar. Cells with reduced *ATG12* expression produced significantly more colonies in soft agar than control cells, similar to *CDKN1A* knockdown, consistent with a role of autophagy in tumor suppression (*Qu et al., 2003*; *Takamura et al., 2011*; *Yue et al., 2003*) (*Figure 6C* and *Figure 6—figure supplement 3*). We observed that *ATG12* knockdown cells have elevated CDKN1A protein expression (*Figure 6—figure supplement 4*, lane 1 and 2), which may trigger the suppression of colony formation and could explain why loss of *ATG12* did not generate as many colonies as loss of *TGM2* (*Figure 6C*). However, the simultaneous knockdown of *ATG12* and *CDKN1A* led to substantially more colonies than a reduction in the expression of each gene individually (*Figure 6C*, column 3 to 5, and *Figure 6—figure supplement 3*), indicating a synergistic effect between *ATG12* and *CDKN1A* in preventing cell transformation, similar to the synergistic effect observed between *TGM2* and *CDKN1A* (*Figure 6A*). This synergy was comparable to the effect of loss of *TP53* expression (*Figure 6C*, column 5 and 6). To further confirm that inhibiting autophagy phenocopies *TGM2* knockdown in colony formation, we performed similar RNAi experiments to analyze two additional autophagy regulator genes, *ATG5* and *BECN1* (known as Beclin 1). ATG5 functions in autophagosome elongation (*Kimmelman, 2011*) and BECN1 functions autophagosome nucleation at the initiation of autophagy (*Kimmelman, 2011*). Knockdown of *ATG5*, *BECN1,* or *ATG12* resulted in the accumulation of LC3-I and a decrease in the ratio of LC3-II/LC3-I (*Figure 6—figure supplement 5*), consistent with a defect in the conversion of LC3-I to LC3-II and a block in the early stage of autophagy, as expected (*Liu et al., 2012*; *Mizushima and Yoshimori, 2007*; *Otomo et al., 2013*; *Papandreou et al., 2008*; *Tang et al., 2013*; *Thorburn et al., 2014*). We found that cells with reduced *ATG5* or *BECN1* expression produced significantly more colonies in soft agar than control cells, recapitulating our observations with *ATG12* knockdown cells (*Figure 6D*, column 1 to 5, and *Figure 6—figure supplements 6* and *7*). Furthermore, the simultaneous knockdown of *ATG5* or *BECN1* with *CDKN1A* led to substantially more colonies than reduced expression of each gene individually (*Figure 6D*). Indeed, the cells with the double knockdown produced the comparable number of colonies to *TP53* knockdown cells (*Figure 6D*, column 7 to 11). These results suggest that the inhibition of autophagy, together with loss of *CDKN1A* expression, strongly promotes transformation in HMECs.

In order to determine if loss of *TGM2* stimulates colony formation primarily through inhibition of autophagy, we expressed *TGM2* shRNA and *ATG12* shRNA in HMEC$^{TERT/ST/ER-RasV12}$ cells. We observed no additional stimulation of colony formation by simultaneous loss of *TGM2/ATG12* compared to single loss of *TGM2* (*Figure 6E* and *Figure 6—figure supplements 8* and *9*). We also coexpressed *TGM2* shRNA with *ATG5* shRNA or *BECN1* shRNA in HMEC$^{TERT/ST/ER-RasV12}$ cells. Similar to the double knockdown of *TGM2/ATG12*, there was no significant additional stimulation of colony formation by simultaneous loss of *TGM2/ATG5* or *TGM2/BECN1* compared to single loss of *TGM2* (*Figure 6F*, column 6 to 10, *Figure 6—figure supplements 10* and *11*). These data suggest that knockdown of *TGM2* promote colony formation possibly by inhibiting autophagy.

Taken together, our data suggest that efficient autophagic flux through autophagic protein degradation and autolysosome clearance by TGM2, together with cell cycle arrest by CDKN1A, are complementary barriers for tumor suppression in the TP53 pathway, and that simultaneous loss of these barriers is important for oncogenic transformation in HMECs.

## The GTPase activity of TGM2 is important to promote autophagic flux

TGM2 is a multifunctional enzyme with two well-established activities: crosslinking as a transglutaminase and binding and hydrolyzing GTP as a GTPase (*Chhabra et al., 2009*; *Lorand and Graham, 2003*). To investigate which of these functions are important to promote autophagic protein degradation and autolysosome clearance, we treated cells with the TGM2 inhibitors Z-DON and LDN 27219. Z-DON is a peptide-based inhibitor that specifically inhibits the crosslinking activity of TGM2 (*McConoughey et al., 2010*), whereas LDN 27219 inhibits the GTPase activity of TGM2 (*Case and Stein, 2007*). HMEC[TERT/ST/ER-RasV12] cells were deprived of growth supplements, with or without TGM2 inhibitors, and the levels of SQSTM1 and LC3-I and -II were assessed. Interestingly, treatment with LDN 27219, but not Z-DON, led to a significant accumulation of SQSTM1 protein (*Figure 6G*). Furthermore, LC3-II protein levels and the ratio of LC3-II/LC3-I were increased dramatically in LDN 27219 treated cells, and partially in Z-DON treated cells (*Figure 6G*), suggesting that the GTPase activity of TGM2 is more important than its crosslinking activity for promoting autophagic protein degradation.

## The GTPase activity of TGM2 contributes to the suppression of cell transformation

We showed that the GTPase activity of TGM2 is important for promoting autophagic protein degradation (*Figure 6G*). Therefore, we hypothesized that colony formation will be stimulated in soft agar if the GTPase activity of TGM2 is inhibited. Instead of using TGM2 inhibitors, which may not be effective in agar and may have a side effect by prolonged culture during colony formation, we evaluated *TGM2* mutants that alter the critical amino acid residues for GTP binding (R580A) or for transamidation (C277S) (*Gundemir et al., 2012*; *Kumar et al., 2012*; *Kumar and Mehta, 2012*). We expressed wild-type or mutant *TGM2* constructs in *TGM2* knockdown cells. Consistent with *Figure 1F*, ectopic expression of wild-type TGM2 suppressed the colony formation in *TGM2* knockdown cells (*Figure 6H*, column 1 to 3). Interestingly, the GTP binding site mutant (R580A) completely ablated the tumor suppressive effect of TGM2 in colony formation, whereas the transamidation site mutant (C277S) displayed only a slightly reduced tumor suppressive effect compared to wild-type TGM2 (*Figure 6H* and *Figure 6—figure supplement 12*). The comparable expression levels of endogenous TGM2, and the shRNA-resistant wild-type and mutant TGM2 are shown in *Figure 6—figure supplement 13*. These findings are consistent with the observation that the GTPase inhibitor prevented autophagic protein degradation (*Figure 6G*). Thus, the GTPase activity of TGM2 is required to promote autophagy and suppress cell transformation of HMEC[TERT/ST/ER-RasV12] cells.

## Discussion

Our loss-of-function screen identified *TGM2* as a putative tumor suppressor gene within the TP53 signaling pathway that prevents oncogenic transformation and tumor formation by primary HMECs expressing TERT, activated HRAS[V12] and SV40 small T antigen. The role of TGM2 in cancer is quite complex and remains poorly understood. *TGM2* has been shown to induce apoptosis (*Fok and Mehta, 2007*) or differentiation (*Liu et al., 2007*), and inhibit angiogenesis (*Jones et al., 2006*). Additionally, the *TGM2* gene locus is epigenetically silenced via methylation in some breast tumors and gliomas (*Ai et al., 2008*; *Dyer et al., 2011*). In contrast, *TGM2* is overexpressed in other types of tumors (*Iacobuzio-Donahue et al., 2003*; *Jin et al., 2012*; *Miyoshi et al., 2010*). *TGM2* was also reported to be upregulated in cancer cell lines by several important signaling pathways involved in tumor progression or metastasis, such as NFKB1/NF-κB, TGFB1/TGF-beta, RARA/RAR-alpha (*Ai et al., 2012*; *Cao et al., 2012*; *Jung et al., 2007*; *Ranganathan et al., 2007*; *Rebe et al., 2009*), and upon genotoxic stress (*Caccamo et al., 2012*; *Shin et al., 2004*). Although the role of *TGM2* in tumorigenesis is likely context-dependent, our data clearly reveal a tumor suppressive role of TGM2 in a variety of cell lines that represent an early step in transformation and carcinogenesis.

We found that the expression of *TGM2* is dependent on TP53, and that *TGM2* is a potential direct target gene of TP53. Although there are several putative consensus TP53 binding motifs in the *TGM2* promoter, we found that an 82 bp region which does not contain a predicted TP53 binding consensus motif is necessary and sufficient for TP53-mediated transactivation. Our ChIP analysis also showed that endogenous TP53 binds to this region in HMECs. Recent genome-wide approaches have revealed that around 10% of the validated TP53 responsive elements are novel sequences that are not clearly related to the canonical TP53 binding consensus (*Menendez et al., 2009*), underscoring the complexity of the TP53 network (*Contente et al., 2002*; *Jordan et al., 2008*; *Menendez et al., 2013*; *Tebaldi et al., 2015*). The TP53 binding sequence in the *TGM2* promoter could be one of these 10% which do not have a canonical consensus. Our database search did not identify other promising transcription factor candidates that bind to this region. Although TP53-mediated transactivation of a reporter construct containing this region was observed within 24 hr after transfection in TP53-null H1299 cells, induction of *TGM2* mRNA by Nutlin-3a in HMECs and BJ cells required 48 hr. The distinct kinetics of TGM2 induction by TP53 in these different contexts may reflect differences in epigenetics, co-factors, repressors, or posttranscriptional modifications of TP53, which remain to be elucidated.

TGM2 is a pleiotropic enzyme with well-known transglutaminase and GTPase activities (*Lorand and Graham, 2003*). The transamidation activity of TGM2 has been implicated in apoptosis by interacting with BAX (*Rodolfo et al., 2004*) or cross-linking CASP3/Caspase 3 and RB1/pRB to inhibit apoptosis (*Boehm et al., 2002*; *Oliverio et al., 1997*; *Yamaguchi and Wang, 2006*) in various cancer cell lines. On the other hand, the GTP/GDP binding but not the transamidation domain of TGM2 has been shown to function in the epithelial-to-mesenchymal transition in immortalized MCF10A cells (*Mann et al., 2006*). Our data suggest that the GTPase function of TGM2 is required for autophagy and suppresses transformation of HMEC$^{TERT/ST/ER-RasV12}$ cells. It is possible that the cell type, *TP53* status or cell culture conditions influence the biochemical activities of TGM2; for instance, high $Ca^{2+}$ concentrations induce TGM2 transamidation activity but inhibit its GTPase activity (*Chhabra et al., 2009*; *Lorand and Graham, 2003*). The predominant activity of TGM2 in specific contexts may determine whether it functions as a tumor-promoting or -suppressive protein (*Chhabra et al., 2009*).

Similarly, the role of autophagy in cancer is quite complex, and may have tumor suppressive or promoting effects depending on the model and stage of tumorigenesis. Therefore, its role in cancer must be determined for each context. Substantial evidence suggests that autophagy supports the survival of established tumors by providing nutrients under metabolic stress. Alternatively, autophagy can act as a tumor suppressor by enhancing the degradation of damaged proteins and organelles to maintain tissue homeostasis and genomic stability in normal cells or in the early stages of cancer development (*Green and Levine, 2014*; *Lorin et al., 2013*; *Mizushima and Komatsu, 2011*). Tumor suppressive roles for autophagy were demonstrated in mice with *Becn1/Beclin*-1 heterozygosity, systemic mosaic *Atg5* deletion or liver-specific deletion of *Atg7* (*Green and Levine, 2014*; *Lorin et al., 2013*; *Mizushima and Komatsu, 2011*). Consistent with a tumor suppressive role, we found that TGM2 promotes autophagy and prevents an early step of HMEC transformation, the acquisition of anchorage-independent growth.

Our data suggest that TGM2 enhances autophagic protein degradation and autolysosome clearance, thereby promoting autophagic flux. Previous work described a role for TGM2 in autophagosome maturation (*D'Eletto et al., 2009*). These findings suggest that TGM2 is an important regulator of autophagy. Although our HMEC transformation model suggests that TGM2-mediated autophagy suppresses early events during tumor initiation, the autophagic function of TGM2 may promote tumor progression by facilitating the survival of established tumors under nutrient stress. In fact, *TGM2* is overexpressed in subset of tumors (*Iacobuzio-Donahue et al., 2003*; *Jin et al., 2012*; *Miyoshi et al., 2010*). TGM2 has been also reported to stimulate epithelial-to-mesenchymal transition (EMT) and remodel the extracellular matrix (*Kotsakis and Griffin, 2007*), supporting a positive role of TGM2 in the later stages of tumorigenesis.

In our model, TGM2 contributes to a TP53-induced autophagy program and suppress transformation; however, TP53 has diverse roles in autophagy. Nuclear TP53 promotes autophagy through many of its target genes, such as *DRAM1, C12orf5/TIGAR, DAPK1, SESN2/SESTRIN2, ULK1, ULK2, BBC3/PUMA, BAX, BAD*, and *BNIP3* (*Balaburski et al., 2010*; *Berkers et al., 2013*; *Itahana and Pervaiz, 2014*; *Levine and Abrams, 2008*). On the other hand, cytoplasmic TP53 and mutant TP53

inhibit autophagy (*Berkers et al., 2013*; *Itahana and Pervaiz, 2014*). Therefore, the role of TP53 in autophagy must be determined in each model system and cell context. We observed that TP53 promotes autophagic flux and autophagosome formation, an early step of autophagy, in HMECs. However, these findings do not exclude a role for TP53 in later steps of autophagy through TGM2.

Interestingly, we found that TGM2 and CDKN1A provide complementary functional contributions to tumor suppression in the TP53 pathway. Furthermore, knockdown of core autophagy genes (*ATG12, ATG5*, and *BECN1*) synergized with loss of *CDKN1A* but not with loss of *TGM2* to induce colony formation. In addition, inhibition of the GTPase activity of TGM2 prevents autophagic protein degradation as well as colony formation, supporting the conclusion that TGM2 contributes to tumor suppression, at least in part, by promoting autophagy. Although TGM2 may also have non-autophagic functions to suppress the transformation, our study suggests that cell cycle arrest, mediated by CDKN1A, and autophagy, mediated by TGM2, are two critical TP53-dependent tumor suppressive barriers that prevent oncogenic transformation of HMECs (*Figure 7*).

Knockdown of multiple genes by shRNAs can potentially lead to a synergistic effect, even if genes work in the same pathway, due to the incomplete loss of transcripts. For example, we observed that the combined knockdown of *TP53* and *TGM2*, using *TGM2#1* shRNA, produced a greater number of colonies compared to *TP53* knockdown alone. However, we do not exclude the possibility of TP53-independent functions of TGM2 in suppressing colony formation. Further work will be necessary to elucidate other tumor suppressive functions and regulation of TGM2.

The canonical functions of TP53 are the induction of cell cycle arrest, senescence, and apoptosis upon cellular stress. However, recent evidence challenges this long held view of TP53-mediated tumor suppression and highlight the importance of non-canonical, diverse functions for TP53 such as in autophagy (*Bieging and Attardi, 2012*; *Brady et al., 2011*; *Li et al., 2012*; *Valente et al., 2013*). In this manuscript, we provide evidence that TGM2 suppresses an early event in tumorigenesis, anchorage-independent growth, and participates in TP53-induced autophagy which can collaborate with CDKN1A-mediated cell cycle arrest, the canonical tumor suppressive function of TP53 (*Figure 7*). We showed that *TGM2* is a potential direct target gene of TP53 and revealed a role of TGM2 in suppressing colony formation by promoting autophagic flux through autophagic protein

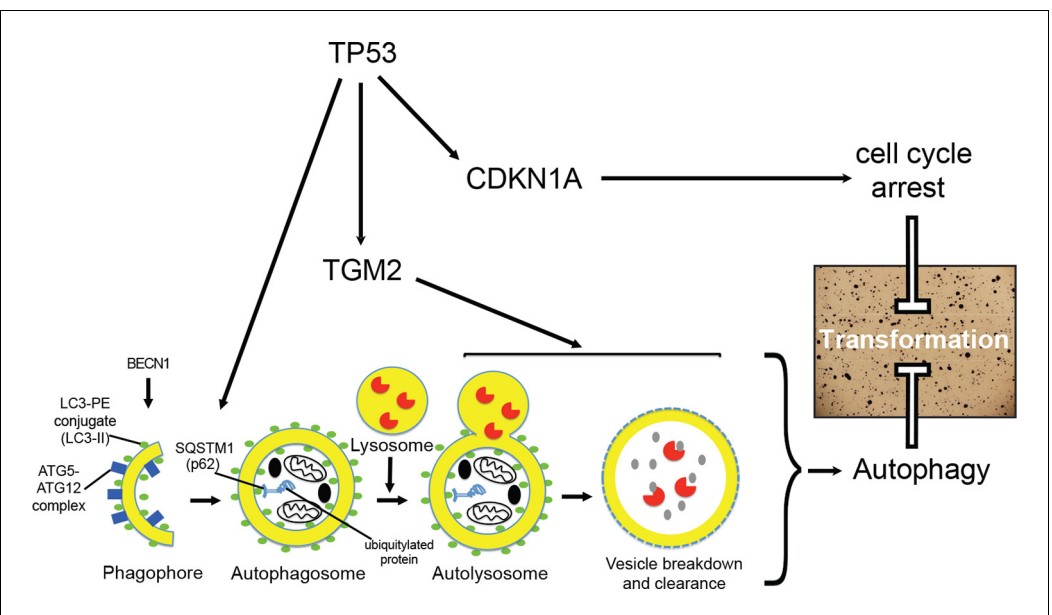

**Figure 7.** A model of the tumor suppressive functions of TGM2 in HMEC[TERT/ST/ER-RasV12] cells. Stress, in this case the depletion of growth supplements, induces autophagy and TP53-dependent expression of *TGM2*. TGM2 facilitates autophagic flux by promoting autophagic protein degradation and autolysosome clearance. Loss of *TGM2* expression synergizes with loss of *CDKN1A* expression to promote malignant transformation of HMECs. Therefore, TGM2-mediated autophagy and CDKN1A-mediated cell cycle arrest are potentially two critical barriers in the TP53 pathway that prevent oncogenic transformation of HMECs.

degradation and autolysosome clearance. These findings are consistent with a recent report showing that induction of autophagy is part of the TP53 tumor suppressive response during early tumorigenesis (*Kenzelmann Broz et al., 2013*). Therefore, inhibition of autophagy as a therapeutic strategy for cancer may have unintended, tumorigenic effects in cases where autophagy is a critical part of the early tumor suppressive response. Weakening the TP53-dependent tumor suppressive barrier by inhibiting autophagy may allow early lesions with low level oncogenic signaling to progress to more aggressive lesions (*Junttila et al., 2010*). This caveat needs to be considered before autophagy inhibition is used for cancer therapy in the clinic.

## Materials and methods

### Candidate selection and screen

The procedure to select gene candidates for new TP53 pathway components was previously described (*Drost et al., 2010*). Briefly, the candidates were selected from the Miller breast expression array (GSE3494) consisting of 251 breast cancer samples with *TP53* mutation status (*Miller et al., 2005*). Normalized mRNA signals from gene probes were arranged as lowest = 0, highest = 1. By comparing *TP53* wild-type tumors with *TP53* mutant samples, we selected 122 gene candidates with significant downregulation (p<0.01) in gene signal in a subset of *TP53* wild-type tumors using Chi-square ($\chi$2) analysis. The signal level cut-off is indicated at average minus one standard deviation (horizontal line). For each gene, we constructed a shRNA using the RNAi consortium library database (http://www.broadinstitute.org/rnai/trc/lib), retrovirally transduced it into HMEC-TERT/ST/ER-RasV12 cells, and evaluated the effect of knockdown on colony formation in soft agar.

### Antibodies

Antibodies were obtained from: GeneTex, Inc (TGM2, #GTX111702), Cell Signaling (CDKN1A, #2947; LC3B, #2775; ATG12, #D88H11; ATG5, #D5F5U; BECN1, #D40C5), Santa Cruz Biotechnology (TP53, DO-1, #sc-261), Millipore ($\beta$-actin, #MAB1501), and Novus Biologicals (SQSTM1/p62, 2C11, #H00008878-M01).

### Constructs

The RFP-GFP-LC3 plasmid was a gift from T. Yoshimori (Osaka University, Osaka, Japan) (*Kamimoto et al., 2006*). Retroviral vectors expressing oncogenic SV40 small T antigen, HRAS$^{V12}$, ER-HRAS$^{V12}$, and TERT, and pRetroSuper vectors targeting *TP53, CDKN1A,* and *p16$^{INK4a}$* were described previously (*Mullenders et al., 2009*; *Voorhoeve and Agami, 2003*; *Voorhoeve et al., 2006*). The retroviral pRetroSuper vector expressing mCherry with various shRNA was constructed by replacing the blasticidin marker with mCherry. The retroviral vector expressing TERT-H2B-GFP was sub-cloned into pBabe vector to express TERT with H2B-GFP as selection marker (*Kolfschoten et al., 2005*). pRetroSuper vectors targeting *TGM2* (*TGM2*#1; 5'-ACAGCAACCTTC TCATCGAGT, and *TGM2*#2; 5'-CCACCCACCATATTGTTTGAT), *ATG12* (5'-TGTTGCAGCTTCCTAC TTCAA-3'), *ATG5* (*ATG5*#1; ATTCCATGAGTTTCCGATTGATGGC, and *ATG5*#2; CCTTTGGCCTAA-GAAGAAA), *BECN1* (*BECN1*#1; GATACCGACTTGTTCCTTA, and *BECN1*#2; CTAAGGAGCTGCCG TTATA) and mouse *Tgm2* (*Tgm2*#1; GCTGGACCAACAGGACAATGT, and *Tgm2*#2; GCGAGATGA TCTGGAACTTCC) (*Lock et al., 2011*) were driven by a U6 promoter. The double knockdown vectors of *TGM2* and *CDKN1A* were created by cloning the *CDKN1A* shRNA expression cassette into the pRetroSuper shRNA vector targeting *TGM2* (hairpins #1 and #2). The human *TGM2* ORFs were cloned into miR-Vec expression vectors (*Voorhoeve et al., 2006*). The *TGM2$^R$* ORF contains five synonymous changes in nucleotides 931–939 of transcript variant 1 (CT<u>T</u>CT<u>C</u>ATC to <u>T</u>T<u>G</u>TT<u>G</u>AT<u>T</u>). The human *TP53* ORF was cloned into the pMSCV blast vector together with a 6x His-tag at the C-terminus. The *TP53* construct designed to resist *TP53* shRNA knockdown contains four synonymous changes in nucleotides 983–988 of transcript variant 1 (A<u>G</u>TGG<u>T</u>AA to <u>T</u>CCGG<u>A</u>AA) that preserve the amino acid sequence. TGM2 C277S and R580A mutants were generated by site-directed mutagenesis using *TGM2* wild-type ORF as the template. All constructs were verified by sequencing.

## Cell culture

Primary human mammary epithelial cells (HMECs) (Lot no. 7F3286, #CC-2551, Lonza) were cultured in MEGM media supplemented with Bullet kit containing EGF, insulin, hydrocortisone, bovine pituitary extract, and GA-1000 (gentamicin and amphotericin) as recommended by the manufacturer, and transduced to express the ecotropic receptor and TERT-H2B-GFP as described (*Voorhoeve and Agami, 2003*), selected, and frozen down as early passages. Upon analysis, most of these cells showed a normal karyotype, and 3/13 cells analyzed had trisomy for Chromosome 20. Fully transformed HMECs recovered as a colony from soft agar were analyzed by spectral karyotyping (SKY) and confirmed to have a normal karyotype with no translocations. HMECs that were retrovirally transduced to express TERT-H2B-GFP, SV40 small T, and ER-HRAS$^{V12}$ were referred to as HMEC-$^{TERT/ST/ER-RasV12}$ cells. For deprivation of growth supplements, EGF, insulin, and hydrocortisone were removed from the medium for 24 hr after 3 days of treatment with 300 nM of 4-OHT (#H6278, Sigma). HEK293T cells, human foreskin fibroblast BJ cells and mouse embryonic fibroblast NIH 3T3 cells (ATCC) were grown in Dulbecco's modified Eagle medium supplemented with 10% FCS and antibiotics. Early-passage wild-type and *Tp53*$^{-/-}$ mouse embryonic fibroblasts (MEFs) previously described (*Itahana et al., 2007*; *Itahana and Zhang, 2008*) were kindly provided by Dr. Yanping Zhang (UNC, Chapel Hill, NC), and cultured similarly. All cells were maintained in a 5% CO2 incubator at 37°C. Nutlin-3a was purchased from Sigma (#SML0580) for TP53 activation. For the treatment of TGM2 inhibitor, HMEC$^{TERT/ST/ER-RasV12}$ cells were incubated with 300 nM of 4-OHT together with Z-DON (#Z006, Zedira, 50 µM) or LDN 27219 (#4602, Tocris Bioscience, 10 µM) for 2 days, followed by the deprivation of growth supplements for 24 hr in the presence of the inhibitors.

## Viral transduction

Ecotropic retroviruses were generated as previously described (*Brummelkamp et al., 2002*). Briefly, cells at 60–70% confluence were transduced overnight in the presence of 8 µg/ml polybrene with ecotropic retroviruses (*Brummelkamp et al., 2002*) and selected with blasticidin (4 µg/ml), puromycin (2 µg/ml), or hygromycin (300 µg/mL) 48 hr after transduction.

## Transfection

HMEC$^{TERT/ST/ER-RasV12}$ cells were transfected with plasmids for 4 hr with JetPrime Polyplus (Bioparc, France) following the manufacturer's instructions.

## Generation of *TP53*$^{-/-}$ clones

*TP53* knockout cells were generated by first transfecting HMEC$^{TERT/ST/ER-RasV12}$ cells with plasmids expressing Cas9 and a gRNA (Addgene #41815 and #41824) (*Mali et al., 2013*) targeting exon 4 of *TP53* (GGCAGCTACGGTTTCCGTCT) followed by plating in soft agar and picking of single clones after two weeks of incubation (*Ho et al., 2013*). Single cell clones were expanded, followed by DNA extraction. DNA sequencing was performed to verify frame-shifting mutations in both alleles of *TP53* (*Figure 4—figure supplement 1*).

## Soft agar assay

A bottom layer of 2 ml 1% agar was overlaid with a suspension of 20,000 HMECs in 2 ml 0.4% low gelling agarose (#A9045, Sigma) in DMEM with 10% FCS. Agar was topped up with 1 ml DMEM containing 500 nM of 4-OHT for the first 3 days, and then replaced with DMEM containing EGF (5 ng/ml), insulin (5 µg/ml), hydrocortisone (500 ng/ml), and 4-OHT (500 nM). The top media was replaced after one week. After 2 weeks, colonies were stained with 500 µl of 5 mg/ml MTT (#5655, Sigma) at 37°C for 1 hr and photographed with a stereomicroscope (SZX16, Olympus). Counting of colonies was automated using a MATLAB script based on the intensity and size of stained colonies. Statistical analysis of data was done by unpaired student's two-sided *t*-test. For BJ cells, a bottom layer of 2 ml of 1% agar was overlaid with a suspension of 40,000 BJ cells in 2 ml of 0.4% low gelling agarose. Agar was topped up with 1 ml DMEM containing 500 nM of 4-OHT and 10% FCS for the first 1 week. After 1 week, the top medium was replaced with DMEM containing EGF (5 ng/ml), insulin (5 µg/ml), hydrocortisone (500 ng/ml), 4-OHT (500 nM), and 10% FCS. For NIH 3T3 cells, a bottom layer of 2 ml of 1% agar was overlaid with a suspension of 20,000 NIH 3T3 cells in 2 ml of 0.4% low gelling agarose. Agar was topped up with 1 ml DMEM containing 500 nM of 4-OHT and 10%

FCS. The top media was refreshed after one week. After 2 weeks, colonies were stained with MTT and analyzed in the same way as HMEC colonies.

## Quantitative RT-PCR

RNA was extracted using Trizol (Life Technologies) and 1 µg of total RNA was used for cDNA synthesis (iScript, Biorad). RT-qPCR was performed using SsoFast EvaGreen supermix (Biorad) on a CFX96 machine (Biorad, CA, USA) according to manufacturer's instructions. *TBP* or *Gapdh* were used as an internal control. Statistical analysis of data was done by unpaired student's two-sided *t*-test. Specific qPCR primers (5' to 3'):

*TGM2* forward: ACTACAACTCGGCCCATGAC
*TGM2* reverse: TGGTCATCCACGACTCCAC
*TP53* forward: CAACAACACCAGCTCCTCTC
*TP53* reverse: CCTCATTCAGCTCTCGGAAC
*CDKN1A* forward: GCAGACCAGCATGACAGATTT
*CDKNA1* reverse: GGATTAGGGCTTCCTCTTGGA
*MDM2* forward: GAATCTACAGGGACGCCATC
*MDM2* reverse: TCCTGATCCAACCAATCACC
*TBP* forward: TCCTGTGCACACCATTTTCC
*TBP* reverse: CGCCGAATATAATCCCAAGC
Mouse *Tgm2* forward: GGCCACTTCATCCTGCTCTA
Mouse *Tgm2* reverse: TCCAAGGCACACTCTTGATG
Mouse *Cdkn1A* forward: CCTGGTGATGTCCGACCTG
Mouse *Cdkn1A* reverse: CCATGAGCGCATCGCAATC
Mouse *Gapdh* forward: AGGTCGGTGTGAACGGATTTG
Mouse *Gapdh* reverse: TGTAGACCATGTAGTTGAGGTCA

## Western blot

Cells were washed once with Phosphate Buffered Saline (PBS) and lysed with 2% SDS lysis buffer (2% SDS, 50 mM Tris-HCl [pH 6.8], 10% glycerol). Protein concentration was determined with the BCA protein assay kit (Thermo Scientific, MA, USA). Equal amounts of protein were separated by SDS-polyacrylamide gel electrophoresis and transferred to PDVF membranes. Membranes were blocked with 4% non-fat milk (Biorad, CA, USA) and incubated with the indicated antibodies. Detection of blots was done with Western Lightning Plus-ECL reagent (PerkinElmer, MA, USA) for antibody conjugated with HRP, or done with Odyssey Infrared Imaging system (LI-COR Biosciences) for fluorescent-labeled antibody.

## Luciferase reporter assays

The *TGM2* promoter from -5980 to -78 base pairs upstream of the translational start site (ATG) were amplified by PCR using the genomic DNA from HMEC[TERT/ST/ER-RasV12] cells as the template, and cloned into the pGL4.11-Luc reporter plasmid (Promega). Deletions of this genomic fragment were also amplified by PCR and cloned into the same reporter plasmids. pGL4.11-Luc reporter plasmid containing *TGM2* genomic fragments were transfected into H1299 cells (*TP53*-null) along with CMV-TP53 and pRL-CMV Plasmid. The luciferase and renilla luciferase activities were measured 24 hr after transfection using a dual-luciferase reporter assay system (Promega) (*Itahana et al., 2015*). Renilla luciferase activity was used as an internal control to normalize transfection efficiency.

## Confocal Imaging and quantification of puncta size

HMEC[TERT/ST/ER-RasV12] cells were seeded onto µ-plates (ibidi, Martinsried, Germany) before transfecting with the RFP-GFP-LC3 construct. EGF, insulin, and hydrocortisone were removed 24 hr post-transfection to stimulate autophagy. After 24 hr, the autophagic vesicles labeled with RFP-GFP-LC3 were then acquired using a 561 nm and a 488 nm lasers on a confocal microscope (Carl Zeiss LSM 710, Jena, Germany) equipped with oil-immersion objective lens (NA 1.40, 63x; Plan Apochromat, Carl Zeiss) and ZEN 2010 software (version 6.0.0.485; Carl Zeiss). The size of the vesicle was obtained from acquired 8-bit images of cells using Image J software (version 1.45f; National Institutes of Health). Briefly, a minimum and a maximum threshold value of 0 and 42, respectively, were

applied to each single cell images. Background noise of 1 pixel was removed before applying the watershed function to obtain the outlines of the vesicles. The size of vesicle, in $\mu m^2$, was obtained using the analyze particles function with the following parameters: size range of 0.2 to 11.0 $\mu m^2$ and circularity of 0.7 to 1.0. Subsequently, the average area of each vesicle per cell was represented as box plot generated using the GraphPad Prism 5 software (version 5.03). Statistical analysis of data was done by unpaired student's two-sided $t$-test.

### Transmission electron microscopy

HMEC[TERT/ST/ER-RasV12] cells were seeded onto 4-chambered coverglass (Lab-Tek Chambered Coverglass System) (Nalgene-Nunc, Rochester, NY, USA) and incubated in medium without EGF, insulin, and hydrocortisone for 24 hr to stimulate autophagy. Samples were fixed in 2.5% glutaraldehyde in PBS at 4°C for 1 hr before osmication with 1% osmium tetroxide, pH7.4 for 1 hr. Subsequently the samples were dehydrated through an ascending series of ethanol at room temperature before infiltration with acetone and resin, followed by final embedding in resin which was allowed to polymerise at 60°C for 24 hr. Samples were cut by an ultra-microtome (Leica), mounted on formvar-coated copper grids and stained with lead citrate. The grids were viewed in a JEOL JEM 1010 transmission electron microscopy (TEM).

### Chromatin immunoprecipitation (ChIP) Analysis

Chromatin Immunoprecipitation assay was performed as previously described (*Itahana et al., 2015*). Briefly, HMEC[TERT/ST/ER-RasV12] cells were fixed and lysed. The protein lysates were then sonicated and immunoprecipitated with mouse anti-p53 antibody (Santa Cruz, DO-1) or mouse IgG as the negative control. The bound DNA in the immunocomplex was eluted and used as the template for PCR. PCR primers used are (5' to 3');

*TGM2* forward: TGGGCTAGTTGTGTGTCCCTGTCC
*TGM2* reverse: AGGCGGAGAGCGGCGCTAACTTAT
*CDKN1A* forward: GTGGCTCTGATTGGCTTTCTG
*CDKNA1* reverse: CTGAAAACAGGCAGCCCAAG
*GAPDH* forward: GTATTCCCCCAGGTTTACAT
*GAPDH* reverse: TTCTGTCTTCCACTCACTCC

### Subcutaneous tumorigenicity assay

HMECs expressing TERT, SV40 small T antigen, HRAS[V12] and the respective shRNAs were resuspended in 50% PBS and 50% matrigel (#354248, BD Falcon) and 500,000 cells in 100 μl were injected in each flank of immunocompromised NOD/SCID female mice. Tumor growth was monitored every 3–4 days. Mice were sacrificed after 4–6 weeks of injection when tumors were noticeable and less than 2 cm in diameter. All work was done under an approved animal protocol (IACUC#2013/SHS/815).

## Acknowledgements

We thank Dr. Marc Fivaz for the Matlab script. We thank Dr. Mei Wang-Casey for helpful discussion. We thank Dr. Patrick Tan, Dr. Cheong Jit Kong, Dr. Xu Peng, Dr. Hou Aihua, Dr. Ooi Chia Huey, Dr. Maybelline Giam, Dr. Giulia Rancati, Dr. Gaye Saginc, Lee Guan Hwee Bernard, Benjamin Farah, and Sam Tan Jian Chye for providing technical helps and reagents. We thank Angela Andersen, Life Science Editors for editorial assistance. We also thank Dr. Alexandra Pietersen and Dr. Paul Yen for critical reading of the manuscript.

## Additional information

### Funding

| Funder | Grant reference number | Author |
| --- | --- | --- |
| National Medical Research Council | NMRC/GMS/1303/2011 | Koji Itahana |

| National Medical Research Council | NMRC/GMS/1250/2010 | Mathijs Voorhoeve |
|---|---|---|
| National Medical Research Council | NMRC/CBRG/003½013 | Mathijs Voorhoeve Koji Itahana |
| Ministry of Health -Singapore | MOE2013-T2-1-123 | Koji Itahana |
| Duke-NUS core grant | | Mathijs Voorhoeve Koji Itahana |

The funders had no role in study design, data collection and interpretation, or the decision to submit the work for publication.

### Author contributions

SYY, YI, MV, KIt, Conception and design, Acquisition of data, Analysis and interpretation of data, Drafting or revising the article, Contributed unpublished essential data or reagents; AKG, Acquisition of data, Analysis and interpretation of data, Drafting or revising the article, Contributed unpublished essential data or reagents; RH, KIw, Acquisition of data, Analysis and interpretation of data; HTN, YB, KK, YJW, BHB, Acquisition of data, Contributed unpublished essential data or reagents

### Author ORCIDs

Koji Itahana, http://orcid.org/0000-0002-7241-2894

### Ethics

Animal experimentation: This study was performed in strict accordance with the recommendations of the Institutional Animal Care and Use Committee at the SingHealth, Singapore under an approved animal protocol (IACUC#2013/SHS/815).

# Additional files

### Supplementary files

• Supplementary file 1. Primary screen. Gene symbol, % of number of colonies formed in knockdown cells compared to control cells, and shRNA sequences used are shown for 122 of selected target genes. 4 candidate genes identified in primary screen are shown in red. *CDKN1A* and *TP53* shRNAs were used as positive controls and shown in blue.

• Supplementary file 2. Secondary screen. To exclude off-target effects, additional shRNAs against the 4 genes were constructed in a secondary screen. Knockdown of only one, *TGM2*, produced colonies in soft agar with at least 2 independent shRNAs.

### Major datasets

The following previously published datasets were used:

| Author(s) | Year | Dataset title | Dataset URL | Database, license, and accessibility information |
|---|---|---|---|---|
| Miller LD, Smeds J, George J, Vega VB, Vergara L, Ploner A, Pawitan Y, Hall P, Klaar S, Liu ET, Bergh J | 2005 | An expression signature for p53 in breast cancer predicts mutation status, transcriptional effects, and patient survival | http://www.ncbi.nlm.nih.gov/geo/query/acc.cgi?acc=GSE3494 | Publicly available at the NCBI Gene Expression Omnibus (Accession no: GSE3494) |
| Drost J, Mantovani F, Tocco F | 2010 | BRD7 is a candidate tumour suppressor gene required for p53 function | http://www.ncbi.nlm.nih.gov/geo/query/acc.cgi?acc=GSE20076 | Publicly available at the NCBI Gene Expression Omnibus (Accession no: GSE20076) |

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
