## [Decision Letter]

Thank you for sending your work entitled "Transglutaminase 2 (*TGM2*) contributes to a p53-induced autophagy program to prevent oncogenic transformation" for consideration at *eLife*. Your article has been favorably evaluated by Tony Hunter (Senior Editor) and three reviewers, one of whom is a member of our Board of Reviewing Editors.

The Reviewing Editor and the other reviewers discussed their comments before we reached this decision, and the Reviewing Editor has assembled the following comments to help you prepare a revised submission:

The reviewers agreed that this manuscript reports a very interesting dataset revealing an important role for the multifunctional protein transglutaminase 2 (*TGM2*) in p53-dependent tumor suppression in the context of human mammary epithelial cells (HMEC). Using a defined oncogenic HMEC transformation protocol that is sensitive to p53 status, they perform a focused genetic screen to identify genes whose inactivation overcomes p53-dependent tumor suppression. This screen leads to the identification of *TGM2* as a mediator of p53-dependent tumor suppression in this experimental system. They go on to show that: 1) *TGM2* acts downstream of p53, 2) *TGM2* expression is somehow positively regulated by p53, 3) p53 is a mediator of autophagy in response to growth factor deprivation in this system, 4) *TGM2* is a promoter of autophagy in this context acting at late stages of the autophagy cascade, as it is required for autophagic protein degradation and autolysosome clearance, and 5) depletion of *TGM2* and *CDKN1A* (p21) phenocopies p53 depletion in this setting, indicating that *TGM2*-dependent autophagy and p21-dependent cell cycle arrest may account for the bulk of p53-dependent tumor suppression in this paradigm.

There is an ongoing debate in the p53 field about how exactly this transcription factor suppresses tumor growth. More specifically, it is unclear which p53 target genes deliver tumor suppression in various contexts. Recent reports using mostly mouse models have challenged the long held notion that p53 halts tumor development via transactivation of genes involved in cell cycle arrest, senescence and/or apoptosis. However, it is becoming evident that p53 may employ diverse downstream pathways in a context-dependent manner. Within this framework, this manuscript makes an important contribution to this debate by identifying *TGM2* as a mediator of p53-dependent tumor suppression, and by showing synergistic action of this gene and p21, the key mediator of cell cycle arrest. With all this being said, the reviewers agreed that the paper falls short in terms of overall impact and significance as expected for a high impact journal such as *eLife*.

After discussion, the reviewers decided to encourage a submission of a revised manuscript addressing the following major concerns:

1) Lack of mechanistic insight into the *TGM2*-p53 regulatory connection. Although the data are clear in showing that *TGM2* expression is upregulated by p53, it is not clear how this regulation take place. Is *TGM2* a direct p53 target gene or not? The paper will be strongly enhanced by answering this question. There are myriad experimental approaches to define this question. Is the upregulation at the transcriptional or post-transcriptional level? Is *TGM2* induced in response to p53 activation by small molecule inhibitors of the p53-*MDM2* interaction? If so, is this an immediate (likely direct) or delayed effect? Would this upregulation be observed in a promoter-reporter assay? If so, what is the minimal promoter region that mediates transactivation? Is there a p53 response element in this region or is it mediated by other transcription factors? Regardless of whether *TGM2* is a direct or indirect p53 target, this information will be very valuable.

2) Limited applicability. The entire manuscript relies on data generated from a single experimental system (transformed HMECs). Although this system is elegant in the sense that the oncogenic path is defined, is also somewhat artificial and may have limited applicability to human cancer. Given the ongoing debate in the p53 field and the importance of the question investigated (i.e. how does p53 halt tumor growth?), the paper will be significantly improved by defining the prevalence of the key observations and mechanisms described. More specifically, is *TGM2* a mediator of p53-dependent tumor suppression in other cell types or experimental settings? How prevalent is p53-dependent *TGM2* upregulation? As the authors point out TP53 tumor suppressor functions may be different in different cells. This is a critical point and the paper would have more impact if the authors demonstrated that *TGM2* is important in other cases. Numerous other normal cells have been transformed by the same telomerase activation/ RB/TP53 inactivation/RAS activation approach that underlies the experiments here. It would be valuable to test if *TGM2* is always critical or not. If it's not, it presumably means that different cell types have different critical activities and this is only breast cell specific.

3) Unclear involvement of autophagy downstream of *TGM2* function. A key conclusion in the manuscript is that *TGM2* functions as a tumor suppressor by virtue of its ability to regulate autophagy. While the authors convincingly show that *TGM2* regulates autophagy, it is not properly demonstrated that it is the autophagy activity that is responsible for the effect on transformation. The key experiment to address this is that knockdown of *ATG12* mimics the effect of *TGM2* knockdown and cooperates with *CDKN1A* depletion but not knockdown of *TGM2* to increase colony formation. The reviewers considered this to be insufficient to draw the conclusion that regulation of autophagy is tumor suppressive here. *ATG12* is known to have other effects, e.g. to regulate mitochondrial apoptosis by at least two distinct mechanisms different than autophagy (see Rubinstein et al., Mol Cell, 2011 and Radoshevich et al., Cell, 2010). In order to fully support the conclusion that autophagy is important in this setting, the authors need to repeat these experiments targeting at least two other essential autophagy regulators and show they get the same effects (and demonstrate that they are in fact inhibiting autophagy when they do this).

4) Lack of insight into *TGM2* mechanism of action. What is the mechanism through which increased levels of *TGM2*, which is predominantly secreted but may also have intracellular roles, can stimulate autophagy downstream of p53 (e.g. is it acting as a secreted or intracellular protein, is its transglutaminase activity needed, etc.?). To address this issue, the reviewers proposed a set of 'rescue experiments', where shRNA-resistant *TGM2* expression constructs (expressing either wild type *TGM2* or diverse mutants) are employed to 'rescue' the phenotype seen upon *TGM2* depletion. A 'catalytically dead' mutant would address the issue of whether the transglutaminase activity is required. The authors may be able to generate mutants to dissect extra- versus intra-cellular functions. Even if unsuccessful, these experiments (or similar experiments) should be discussed in the manuscript.

---

## [Author Response]

[…] There is an ongoing debate in the p53 field about how exactly this transcription factor suppresses tumor growth. More specifically, it is unclear which p53 target genes deliver tumor suppression in various contexts. Recent reports using mostly mouse models have challenged the long held notion that p53 halts tumor development via transactivation of genes involved in cell cycle arrest, senescence and/or apoptosis. However, it is becoming evident that p53 may employ diverse downstream pathways in a context-dependent manner. Within this framework, this manuscript makes an important contribution to this debate by identifying TGM2 as a mediator of p53-dependent tumor suppression, and by showing synergistic action of this gene and p21, the key mediator of cell cycle arrest. With all this being said, the Reviewers agreed that the paper falls short in terms of overall impact and significance as expected for a high impact journal such as eLife.

We really appreciate that the Reviewing Editor described the important point of our paper by saying that identifying *TGM2* as a mediator of TP53-dependent tumor suppression makes an important contribution to the debate. We agree with the Reviewing Editor that recent reports have challenged the long held view of the canonical functions of TP53 in tumor suppression and highlighted the importance of non-canonical, diverse functions of TP53 such as in autophagy. Our data showed that *TGM2* can be a TP53 target gene that contributes to a TP53-induced autophagy program which can collaborate with p21/*CDKN1A*-mediated cell cycle arrest, the canonical tumor suppressive function of TP53. We stress this point in Discussion section. Thank you very much for your comment.

*After discussion, the reviewers decided to encourage a submission of a revised manuscript addressing the following major concerns: 1) Lack of mechanistic insight into the TGM2-p53 regulatory connection. Although the data are clear in showing that TGM2 expression is upregulated by p53, it is not clear how this regulation take place. Is TGM2 a direct p53 target gene or not? The paper will be strongly enhanced by answering this question. There are myriad experimental approaches to define this question. Is the upregulation at the transcriptional or post-transcriptional level? Is TGM2 induced in response to p53 activation by small molecule inhibitors of the p53-MDM2 interaction? If so, is this an immediate (likely direct) or delayed effect? Would this upregulation be observed in a promoter-reporter assay? If so, what is the minimal promoter region that mediates transactivation? Is there a p53 response element in this region or is it mediated by other transcription factors? Regardless of whether TGM2 is a direct or indirect p53 target, this information will be very valuable.*

We appreciate the Reviewing Editor's comment. As shown in Figure 2 and Figure 4, new Figure 2,Figure 3, the upregulation of *TGM2* by TP53 was observed at the transcriptional level. When the expression of TP53 was depleted (knocked-down) or eliminated (knocked-out), the levels of *TGM2* were reduced at the mRNA level in human mammary epithelial cells (HMECs), BJ human foreskin fibroblasts, and mouse embryonic fibroblasts (MEFs). As the reviewers suggested, we treated HMECs and BJ cells with Nutlin-3a, a small molecule inhibitor of the TP53-MDM2 interaction. Nutlin-3a induced the expression of *TGM2* at the mRNA level, strictly in a TP53-dependent manner (new Figure 3). We next performed luciferase reporter assays using a variety of deletion constructs corresponding to the *TGM2* promoter region from −5980 to −78 base pairs relative to the translational start site (ATG), as the Reviewing Editor suggested (new Figure 3 and Figure 3—figure supplement 1). It has been reported that the *TGM2* promoter contains two predicted TP53 binding sites, although these sites were not tested for TP53 binding and TP53-mediated transactivation (Ai et al., 2012). There are also other potential consensus TP53 binding motifs identified using computer software, p53MH algorithm described previously (Hoh et al., 2002). However, the results from our series of deletion constructs unexpectedly suggest that these predicted TP53 binding motifs in the *TGM2* promoter are not responsible for TP53-mediated transactivation of *TGM2*. Instead, we identified a region within the *TGM2* promoter from −159 to −78 (82 bp), which does not contain a TP53 binding consensus sequence, which was necessary and sufficient for TP53-mediated transactivation in this assay. A small deletion of 5 or 10 nucleotide at the 5’- or 3’- end from this region significantly reduced TP53-mediated transactivation (Figure 3—figure supplement 1). The binding of endogenous TP53 to this region was confirmed by ChIP assay using HMECs (new Figure 3), suggesting that *TGM2* is directly activated by TP53. Recent genome-wide approaches have shown that around 10% of the validated TP53 responsive elements represent novel sequences that are not clearly related to the canonical TP53 binding consensus (Menendez et al., 2009), revealing the complexity of the TP53 network (Contente et al., 2002; Jordan et al., 2008; Menendez et al., 2013; Tebaldi et al., 2015). The TP53 binding sequence in the *TGM2* promoter could be among the 10%, which do not have canonical consensus. Our search of the TRANSFAC database (Wingender et al., 2000) did not uncover other promising transcription factor candidates that bind to this region. Although TP53-mediated transactivation of a reporter construct containing this region, along with a TP53-expressing construct, was observed within 24 hours after transfection in TP53-null H1299 cells, induction of *TGM2* mRNA by Nutlin-3a in HMECs and BJ cells took 48 hours. Epigenetic changes, induction of additional co-factors, reduction of repressors, or posttranscriptional modifications of TP53, co-factors or repressors by TP53 activation may explain these difference in the induction of *TGM2* mRNA, and these possibilities remain to be elucidated.

We modified the Results and Discussion sections based on these findings. We hope these findings are sufficient for our current manuscript.

2) Limited applicability. The entire manuscript relies on data generated from a single experimental system (transformed HMECs). Although this system is elegant in the sense that the oncogenic path is defined, is also somewhat artificial and may have limited applicability to human cancer. Given the ongoing debate in the p53 field and the importance of the question investigated (i.e. how does p53 halt tumor growth?), the paper will be significantly improved by defining the prevalence of the key observations and mechanisms described. More specifically, is TGM2 a mediator of p53-dependent tumor suppression in other cell types or experimental settings? How prevalent is p53-dependent TGM2 upregulation? As the authors point out TP53 tumor suppressor functions may be different in different cells. This is a critical point and the paper would have more impact if the authors demonstrated that TGM2 is important in other cases. Numerous other normal cells have been transformed by the same telomerase activation/ RB/TP53 inactivation/RAS activation approach that underlies the experiments here. It would be valuable to test if TGM2 is always critical or not. If it's not, it presumably means that different cell types have different critical activities and this is only breast cell specific.

We thank the reviewers for these constructive suggestions to improve the paper. Accordingly, we engineered human normal foreskin BJ fibroblasts retrovirally expressing hTERT, small T, ER-RAS and *p16^INK4a^* shRNA. *p16^INK4a^* expression is not silenced in BJ cells, unlike HMECs in which *p16^INK4a^* expression is silenced due to methylation of its promoter. Therefore, we introduced *p16^INK4a^* shRNA into BJ fibroblasts to disrupt the pRB pathway. To examine the regulation of *TGM2* by TP53, we compared the expression of *TGM2* between control and *TP53* knockdown BJ^TERT/ST/ER-RasV12/shp16^ cells. *TGM2* expression was reduced by knockdown of *TP53* in BJ**^TERT/ST/ER-RasV12/shp16^**cells, suggesting that TP53-dependent *TGM2* upregulation was observed not only in HMEC^TERT/ST/ER-RasV12^, but also in human foreskin fibroblast BJ^TERT/ST/ER-RasV12/shp16^ cells (new Figure 2). In addition, Nutlin-3a induced the expression of *TGM2* in BJ^TERT/ST/ER-RasV12/shp16^ cells in a TP53-dependent manner (new Figure 3). We also observed the reduced expression of *Tgm2* in *Tp53* knockout mouse embryonic fibroblasts (MEFs) compared to wild-type MEFs (new Figure 2). These results suggest that TP53-dependent *TGM2* regulation is not restricted to HMEC^TERT/ST/ER-RasV12^ cells.

As the Reviewing Editor suggested, we added soft agar assays of two more cell lines (BJ^TERT/ST/ER-RasV12/shp16^ cells and NIH 3T3^ER-RasV12^ mouse embryonic fibroblasts that contain wild-type TP53), to show that TGM2 has a prevalent tumor suppressive role in colony formation (new Figure 1). NIH 3T3cells are widely used for transformation assays. The number of colonies was significantly increased upon knockdown of *TGM2* compared to the control cells in both BJ^TERT/ST/ER-RasV12/shp16^ cells and NIH 3T3^ER-RasV12^ cells. These findings strengthen our conclusion that TGM2 functions to prevent oncogenic transformation in a variety of contexts, including human mammary epithelial cells (HMECs), human foreskin fibroblasts (BJ) and mouse embryonic fibroblasts (NIH 3T3). We added images of the colony formation and Western blot assays to show knockdown efficiency of *TGM2* (Figure 1—figure supplement 7–Figure 1—figure supplement 9 and Figure 1—figure supplement 12).

3) Unclear involvement of autophagy downstream of TGM2 function. A key conclusion in the manuscript is that TGM2 functions as a tumor suppressor by virtue of its ability to regulate autophagy. While the authors convincingly show that TGM2 regulates autophagy, it is not properly demonstrated that it is the autophagy activity that is responsible for the effect on transformation. The key experiment to address this is that knockdown of ATG12 mimics the effect of TGM2 knockdown and cooperates with CDKN1A depletion but not knockdown of TGM2 to increase colony formation. The reviewers considered this to be insufficient to draw the conclusion that regulation of autophagy is tumor suppressive here. ATG12 is known to have other effects, e.g. to regulate mitochondrial apoptosis by at least two distinct mechanisms different than autophagy (see Rubinstein et al., Mol Cell, 2011 and Radoshevich et al., Cell, 2010). In order to fully support the conclusion that autophagy is important in this setting, the authors need to repeat these experiments targeting at least two other essential autophagy regulators and show they get the same effects (and demonstrate that they are in fact inhibiting autophagy when they do this).

We thank the referees for these constructive suggestions. We agree that it is very important to look at the effects of other autophagy regulators to fully understand the contribution of TGM2-mediated autophagy in preventing colony formation. As the reviewing editors suggested, we evaluated the knockdown of two more autophagy regulators (BECN1/Beclin 1 and ATG5) in the soft agar assay. ATG5 is known to be involved in autophagosome elongation (Kimmelman, 2011) and BECN1 is known to be involved in autophagosome nucleation, the initiation of autophagy (Kimmelman, 2011) Consistent with the results from *ATG12* knockdown, the single knockdown of *BECN1* or *ATG5* increased the colony formation compared to the control (new Figure 6, column 1 to 5). The double knockdown of *BECN1* or *ATG5* together with *CDKN1A* led to substantially more colonies than a reduction in the expression of each gene individually, and the numbers of colonies were comparable to loss of *TP53* expression (new Figure 6), similar to the effect between *TGM2* and *CDKN1A* (Figure 6). On the other hand, double knockdown of *BECN1* or *ATG5* together with *TGM2* did not lead to additional colony formation compared to the single *TGM2* knockdown (new Figure 6, column 6 to 10), which mimics the results of the double knockdown of *ATG12* and *TGM2* (Figure 6).These results provide further support for the conclusion that knockdown of *TGM2* promotes colony formation mainly by inhibiting autophagy. We also confirmed that knockdown of *ATG12, ATG5*, or *BECN1* caused the accumulation of LC3-I and a decrease in the ratio of LC3-II/LC3-I (new Figure 6—figure supplement 5), indicating a defect in the early process of autophagy, as shown previously by the depletion of *ATG5, ATG12, or BECN1* (Liu et al., 2012; Mizushima and Yoshimori, 2007; Otomo et al., 2013; Papandreou et al., 2008; Tang et al., 2013; Thorburn et al., 2014). These findings are consistent with the role of BECN1 in autophagosome nucleation and the role of ATG5 and ATG12 in autophagosome elongation. Taken together, these findings strengthen our conclusion that TGM2 prevents colony formation primarily through promoting autophagy, and that TGM2-mediated autophagy and CDKN1A-mediated cell cycle arrest are two important barriers in the TP53 pathway that prevent oncogenic transformation. We added images of the colony formation and Western blot analysis to show the knockdown efficiency of each gene (Figure 6—figure supplement 6,Figure 6—figure supplement 7,Figure 6—figure supplement 10 and Figure 6—figure supplement 11).

*4) Lack of insight into TGM2 mechanism of action. What is the mechanism through which increased levels of TGM2, which is predominantly secreted but may also have intracellular roles, can stimulate autophagy downstream of p53 (e.g. is it acting as a secreted or intracellular protein, is its transglutaminase activity needed, etc.?). To address this issue, the reviewers proposed a set of 'rescue experiments', where shRNA-resistant TGM2 expression constructs (expressing either wild type TGM2 or diverse mutants) are employed to 'rescue' the phenotype seen upon TGM2 depletion. A 'catalytically dead' mutant would address the issue of whether the transglutaminase activity is required. The authors may be able to generate mutants to dissect extra- versus intra-cellular functions. Even if unsuccessful, these experiments (or similar experiments) should be discussed in the manuscript.*

In response to our request, the Reviewing Editor agreed to remove this comment since similar experiments requested by the referees have been done in the original manuscript (Figure 6 in revised manuscript).